# Light and dark conditions control the nitrous oxide uptake and emission dynamics in a subarctic, nutrient-poor permafrost peatland
Nathalie Ylenia Triches [1,2] ✉, Abdullah Bolek[1], Mirkka Rovamo [3,4], Richard E. Lamprecht [3], Kseniia Ivanova [1], Wasi Hashmi [3], Theresia Yazbeck [1], Nicholas James Eves[1], Dhiraj Paul [3], Anna-Maria Virkkala [5,6], Timo Vesala[2,7], Christina Biasi[3,8], Maija E. Marushchak [3] & Mathias Göckede [1]

As permafrost in the Arctic thaws due to global warming, emissions of nitrous oxide ($N_2O$), a powerful greenhouse gas, may rise—yet its dynamics in permafrost peatlands remain poorly understood. Here we present 1,487 chamber measurements of $N_2O$ fluxes collected over three snow-free seasons in a subarctic thawing permafrost peatland. Measurements were taken under both light (with sunlight) and dark (without sunlight) conditions. Under light conditions, fluxes were mostly positive, indicating net emissions, and increased with plant activity and cooler soil temperatures. Under dark conditions, fluxes were consistently negative, indicating net uptake, and linked to soil moisture and green plant cover. The contrast between light and dark was clear and consistent. Overall, the ecosystem acted as a small but continuous $N_2O$ sink. Yet, localised areas showed strong $N_2O$ production potential. These results highlight the critical role of light and plant-soil interactions in regulating $N_2O$ fluxes, with implications for improving Arctic greenhouse gas budget estimates.

Nitrous oxide ($N_2O$) is a strong greenhouse gas with a long atmospheric lifetime (109 years) and a 298 times stronger global warming potential than the same mass of carbon dioxide ($CO_2$) over a time frame of 100 years, although its concentration in the atmosphere is more than a thousand times lower than that of $CO_2$. Soils are major contributors to the global $N_2O$ budget, both in natural and managed ecosystems[1]. Arctic soils were previously considered to have a negligible impact on the global $N_2O$ budget due to limited mineral nitrogen (N) availability resulting from slow mineralisation in cold conditions, as well as low N deposition. However, recent studies challenged this assumption by reporting high $N_2O$ emissions from organic- and ice-rich permafrost peatlands in the Northern Hemisphere[2–5]. Because the accurate determination of low $N_2O$ fluxes, including $N_2O$ uptake from the atmosphere, is difficult, there has been a research bias towards high-emitting Arctic peatlands[6]. Yet, nutrient-poor sites dominate the Arctic landscape[7], thus low fluxes can be important on an areal basis.

Insights on $N_2O$ fluxes from sites with limited availability of N are therefore needed to improve our understanding of the $N_2O$ budget in the Arctic.

To understand the drivers of $N_2O$ fluxes in Arctic permafrost soils, it is essential to consider the interplay of key environmental and biogeochemical factors that regulate production and consumption processes. $N_2O$ is primarily produced through nitrification (aerobic conversion of ammonia ($NH_4^+$) to nitrite ($NO_2^-$) and nitrate ($NO_3^-$)) and denitrification (anaerobic reduction of $NO_3^-$), both of which are influenced by soil moisture, temperature and the availability of N and carbon (C) substrates[8,9]. In natural Arctic soils, biological nitrogen gas ($N_2$) fixation is the main source of new N input[10], while mineralisation supplies inorganic N for nitrification and denitrification—the dominant but not exclusive pathways for $N_2O$ emissions[8]. Soil moisture plays a critical role by controlling $O_2$ availability: an intermediate water-filled pore space (60–70%) promotes co-occurrence of aerobic and anaerobic microsites, enabling coupled nitrification-

[1]Max Planck Institute for Biogeochemistry, Jena, Germany. [2]Institute for Atmospheric and Earth System Research/Forest Sciences, Faculty of Agriculture and Forestry, University of Helsinki, Helsinki, Finland. [3]Department of Environmental and Biological Sciences, Faculty of Science, Forestry and Technology, University of Eastern Finland, Kuopio, Finland. [4]Business School, Faculty of Social Sciences and Business Studies, University of Eastern Finland, Kuopio, Finland. [5]Woodwell Climate Research Center, Falmouth, MA, USA. [6]Finnish Meteorological Institute, Helsinki, Finland. [7]Institute for Atmospheric and Earth System Research/Physics, Faculty of Science, University of Helsinki, Helsinki, Finland. [8]Institute of Ecology, University of Innsbruck, Innsbruck, Austria. ✉e-mail: ntriches@bgc-jena.mpg.de

denitrification and often resulting in a net $N_2O$ release[11]. Higher soil temperatures generally enhance microbial activity and $N_2O$ emissions when substrates are not limiting, particularly under warming or permafrost thaw conditions that increase mineral N availability[11–13]. Vegetation cover further modulates these processes and $N_2O$ exchange by influencing soil temperature (through shading), moisture (through transpiration), and competition for reactive N[3,11]. Bare peat surfaces, for example, can become hot spots of $N_2O$ emission due to high mineral N availability and lack of plant uptake[4]. There remains a critical need for long-term, high-resolution field observations to disentangle the relative importance of soil moisture, temperature, N availability and vegetation dynamics in driving $N_2O$ fluxes and uptake across Arctic landscapes.

The large majority of $N_2O$ flux measurements in Arctic regions were made with opaque, closed chambers that exclude solar radiation (e.g. photosynthetically active radiation (PAR)) to minimise changes in temperature and humidity during measurements due to chamber closure. This approach has been justified by the assumption that the microbial processes underlying the $N_2O$ are not affected by light. However, $N_2O$ emissions have been commonly reported to show significant diurnal variability in agricultural and forested sites, with highest emissions occurring both during day- or night-time, emphasising the importance of clarifying the PAR–$N_2O$ relationship[14,15]. In High Arctic soils, evidence for light-dependent $N_2O$ fluxes remains mixed. Stewart et al. observed a tendency for soils to shift from net $N_2O$ sources in the dark to sinks under light conditions, although differences were only marginally significant ($p = 0.07$) and strongly modulated by vegetation and soil moisture[16]. In contrast, Li et al. reported significantly higher $N_2O$ emissions under light than dark conditions in tundra soils[17]. Despite using comparable chamber approaches during the growing season, the two studies differ in both the magnitude and direction of the light response. Li et al. attributed enhanced $N_2O$ emissions under light to increased oxygen availability from photosynthesis, stimulating nitrification and coupled nitrification-denitrification in soils and plant tissues, as well as plant-mediated production or transport of $N_2O$ from the soil[17–19]. Together, these findings indicate that solar radiation can influence $N_2O$ fluxes, but that responses are context-dependent and likely governed by interacting plant–soil processes. Interestingly, to date, this phenomenon has not been examined in pristine northern peatlands, including those affected by permafrost, although Arctic research has predominantly targeted well-drained, C- and N-rich permafrost features (palsas and peat plateaus) known for high $N_2O$ emissions[4,6]. As a result, the relationship between $N_2O$ fluxes and PAR still remain poorly understood, particularly in Arctic regions and pristine peatlands in general.

Intertwined with PAR, the $N_2O$–$CO_2$ flux relationship remains poorly understood, despite its importance for global C –N interactions[20]. Across ecosystems, including boreal bogs and fens, ecosystem respiration (ER) was significantly positively correlated to $N_2O$ fluxes and explained 52–84% of $N_2O$ variation, likely due to shared substrates, concurrent microbial processes, and common controlling environmental factors[20]. While $CO_2$ fluxes (i.e. net ecosystem exchange (NEE)) are routinely partitioned into gross primary production (GPP) and ER, $N_2O$ studies are seldom separated into light and dark measurement periods[16]. Therefore, chamber measurements during natural light conditions, as opposed to artificially darkened chambers, which have been the prevalent method in the past, are crucial to fully understand the relationship between PAR, ER/NEE and $N_2O$ fluxes; yet, this is not common practice.

Thawing permafrost peatlands at the southern boundary of permafrost distribution are experiencing drastic changes due to global warming and could serve as early indicators for shifts in similar ecosystems further north[21,22]. These peatlands exacerbate the complexity of $N_2O$ fluxes by creating highly heterogeneous soils, where contrasting micro environments (e.g. aerobic and anaerobic zones) can exist side by side at very small scales, enabling different N transformation processes to occur simultaneously[23]. For example, drier and collapsed mounds may emit large $N_2O$ fluxes, while wetter and waterlogged areas often show low emissions or even $N_2O$ uptake[5,22]. The latter occurs when denitrifying microbes take up atmospheric $N_2O$ as an alternative electron acceptor when oxygen ($O_2$) and $NO_{3–}$ are absent, reducing $N_2O$ to $N_2$[6,24,25]. Despite their potential ecological importance for the global N cycle, Arctic $N_2O$ sinks and are not well captured in current field datasets. This is because the magnitude of these $N_2O$ sinks, as well as $N_2O$ effluxes, can be so low that reliably measuring them has not been possible with previously available methods. As a result, Arctic $N_2O$ sinks and low $N_2O$ effluxes remain poorly quantified and understood.

To advance understanding of Arctic $N_2O$ sinks and low $N_2O$ effluxes, we investigated the drivers of $N_2O$ fluxes in a thawing (sub-) Arctic permafrost peatland in northern Sweden under both light and dark conditions across a dry-to-wet thaw gradient from palsa to bog to fen, representing contrasting nutrient statuses and micro habitats. At our site, palsas were dominated by lichens, shrubs and mosses; we classified them as palsa lichen (PL) when dominated by lichen only, and palsa moss (PM) when vegetated by shrubs and mosses. Bogs and fens were characterised by peat-forming mosses and graminoids. According to the literature above, we expected to find (1) an $N_2O$ sink in the wet parts of the permafrost peatland[6,24,25], (2) significant differences between light and dark measurements of $N_2O$ fluxes[16,17], and (3) higher, but still low, $N_2O$ fluxes during the warmer summer month July compared to May and September due to higher soil temperature[11–13]. To test these hypotheses, we measured $N_2O$ and $CO_2$ fluxes with manual flux chambers and a portable gas analyser, and both in- and excluding PAR by using transparent chambers (light conditions) and a light-reflective tarp (dark conditions)[26]. Our measurements were taken over three years (2022–2024) in the snow-free season ($n = 1487$) between May and September, a period coinciding with the warmest summer in Fennoscandia in 2000 years[27]. We specifically separated our analyses in light and dark flux measurements to identify if flux drivers and patterns differ in varying light conditions.

## Results and discussion

Compared to previous studies, our study has three main advantages: first, with our measurement set-up, we are confident to be able to measure very low $N_2O$ flux rates[26]; second, we conducted seven field campaigns (totalling to approx. five months in the field), providing us with the most extensive dataset of $N_2O$ fluxes measured by the chamber method in Arctic regions ($n = 1462$, both light and dark measurements), and, as a result, third, the largest dataset with $N_2O$ fluxes measured in both *light and dark* conditions, including night measurements. Our measurement setup is comparable with previous studies, but we explicitly tested the impact of light by comparing $N_2O$ fluxes under the presence (light) and absence (dark) of solar radiation (including PAR). For this purpose, we used a transparent chamber during daytime measurements to allow PAR exposure, and conducted additional measurements under dark conditions during the day and at night (Fig. 1). To the best of our knowledge, our study is the first to do in-situ transparent chamber measurements in a pristine (sub-) Arctic permafrost peatland, confirming previous results from High Arctic soils suggesting that the impact of sunlight on $N_2O$ fluxes may be more important in Arctic ecosystems than previously thought[17]. These findings have significant implications for pan-Arctic $N_2O$ budget estimates.

### Dominant $N_2O$ uptake with a clear seasonal peak across micro habitats

In a recent review of Arctic $N_2O$ fluxes, Voigt et al. synthesised that wetlands and dry upland soils were more likely to serve as $N_2O$ sinks, whereas peatlands would likely serve as $N_2O$ sources[6]. However, these assumptions are based on a very limited amount of data, and only include data collected with opaque chambers. In our study, between September 2022 and August 2024, we measured median (25–75 percentiles) $N_2O$ fluxes of $-0.28$ ($-1.24$, $0.68$) $\mu gN_2O\text{-Nm}^{-2}\,h^{-1}$ ($n = 1462$, both light and dark measurements), including one constant hot spot on a palsa with $N_2O$ emissions of $6.08$ ($2.38$, $21.9$) $\mu gN_2O\text{-Nm}^{-2}\,h^{-1}$ ($n = 79$; see last paragraph). When we exclude this hot spot (as done for most of the analyses), $N_2O$ fluxes were $-0.38$ ($-1.32$, $0.49$) $\mu gN_2O\text{-Nm}^{-2}\,h^{-1}$ ($n = 1383$), indicating an $N_2O$ sink (Figs. 2 and S1). In contrast to the review, our flux measurements thus reveal a small sink,

**Fig. 1 | Conceptualisation of the measurement setup, with transparent (light) and opaque (dark) measurements during the day (left side), and transparent but dark measurements during the night (right side).** Upwards arrows indicate $N_2O$ emissions, whereas downward arrows indicate $N_2O$ uptake into the soil. Photo on the right: Fabio Cian, 'ubiquitous anomaly,' CC BY-NC-ND 4.0. Figure created in BioRender.

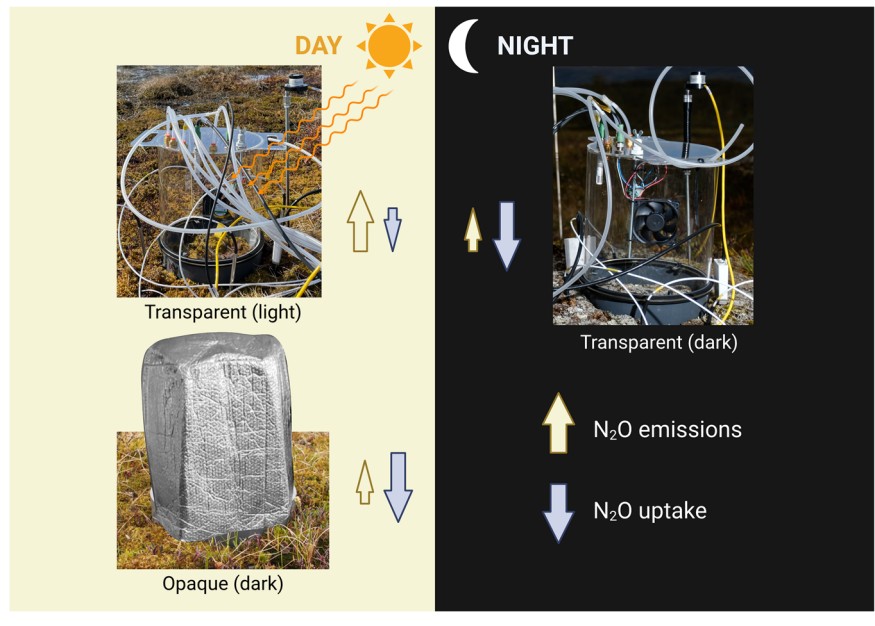

**Fig. 2 | Mean ± SE $N_2O$ fluxes divided into months and micro habitats, excluding one hot spot (light and dark conditions combined), with PL and PM standing for palsa lichen and palsa moss, respectively.** Please note that the measurements in each month may be conducted in various years, with campaigns conducted in May 2023, June 2023, July 2023 and 2024, August 2024 and September 2022 and 2023. Letters indicate significant differences between micro habitats according to ANOVA and Tukey HSD post-hoc tests, with differing letters between micro habitats indicating significant differences. The red dashed horizontal line indicates the border between a source (positive values) and sink (negative values).

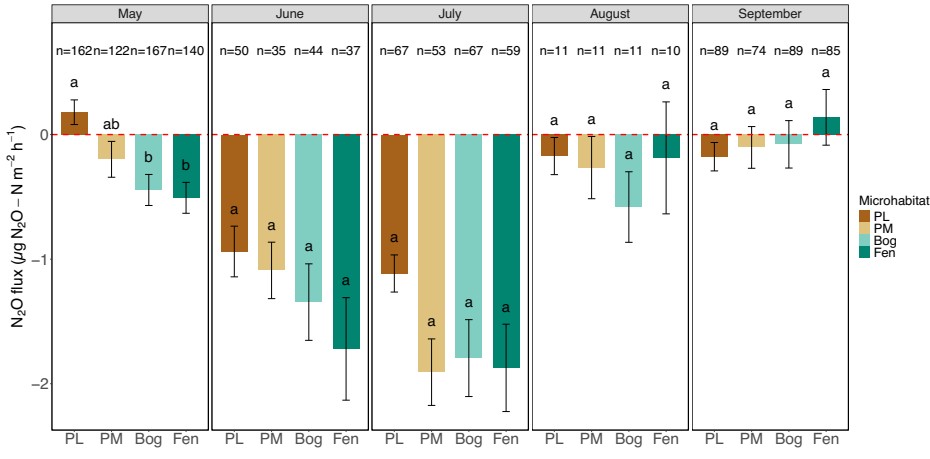

whereas the review reported a source for peatlands (bogs, fens, peat plateaus and palsas) with median (25–75 percentiles) $N_2O$ fluxes of 2.5 (0.75, 20.04) µg $N_2O$-N m$^{-2}$ h$^{-1}$ ($n = 30$)[6]. The 30 peatland observations in the recent review are strongly biased toward high $N_2O$ emissions, as eight originate from bare palsa surfaces where well-drained conditions, uplifted permafrost, and erosion of the ombrotrophic peat expose nutrient-rich deeper peat, while fen and bog sites, and observations at higher water-filled pore space (>60%), are under-represented. Our values were also lower than the previously reported sink for wetlands other than peatlands (i.e. marshes and swamps; median (25–75 percentiles): 0.42 (−0.33, 4.83) µg $N_2O$-N m$^{-2}$ h$^{-1}$ ($n = 25$)), even when we included the hot spot.

We also observed a pronounced seasonal course in our $N_2O$ measurements (Fig. 2), with the sink strength peaking for all micro habitats in the warm summer months June and July. This seasonal pattern could be explained by $NH_4^+$ availability: in fens and bogs, $NH_4^+$ was highest in the early growing season, and then decreased throughout the summer (as seen in Figs. S3 and S4), likely due to N uptake by plants, decreasing the soil mineral N pool. Overall, the vast majority of monthly mean fluxes were negative, except for net emissions observed from PL in May (median (25–75 percentiles) 0.3 (−0.49, 1.00) µg$N_2O$-N m$^{-2}$ h$^{-1}$, $n = 162$), and from fen in September (median (25–75 percentiles) 0.11 (−1.05, 1.15) µg$N_2O$-N m$^{-2}$ h$^{-1}$, $n = 85$).

The discrepancy of the recent review and our results is likely due to methodological differences and the amount of observations considered. Using a portable gas analyser, we employed short chamber closures (5 min compared to the typical 30–60 min)[28], which are critical for detecting $N_2O$ uptake. Under diffusion-limited conditions, $N_2O$ is rapidly depleted from the chamber headspace, so longer closures can underestimate uptake rates[26]. Our brief closure time, therefore, likely enabled detection of the palsa sink that earlier studies may have missed, enhancing the overall $N_2O$ sink observed here. However, since former studies have relied exclusively on opaque chambers, they may underestimate $N_2O$ emissions, especially in shoulder seasons, during which measurements are rarely conducted.

To the best of our knowledge, ours is the first study to report a persistent, albeit small $N_2O$ sink in a (sub-) Arctic permafrost peatland over several years, and the first to observe continuous net uptake on the dry, uplifted palsa surfaces. We can thus extend our hypothesis, to find an $N_2O$ sink in bogs and fens, to dry palsa surfaces (20–32% mean volumetric water content in the top 30 cm depth; see Fig. S2). This phenomenon has previously been reported only for mineral High-Arctic soils[16,17,29] and in a laboratory mesocosm experiment using permafrost peatland soil[23], but not in-situ in (sub-) Arctic permafrost peatlands. While Brummell et al. used a method with a detection limit that rendered

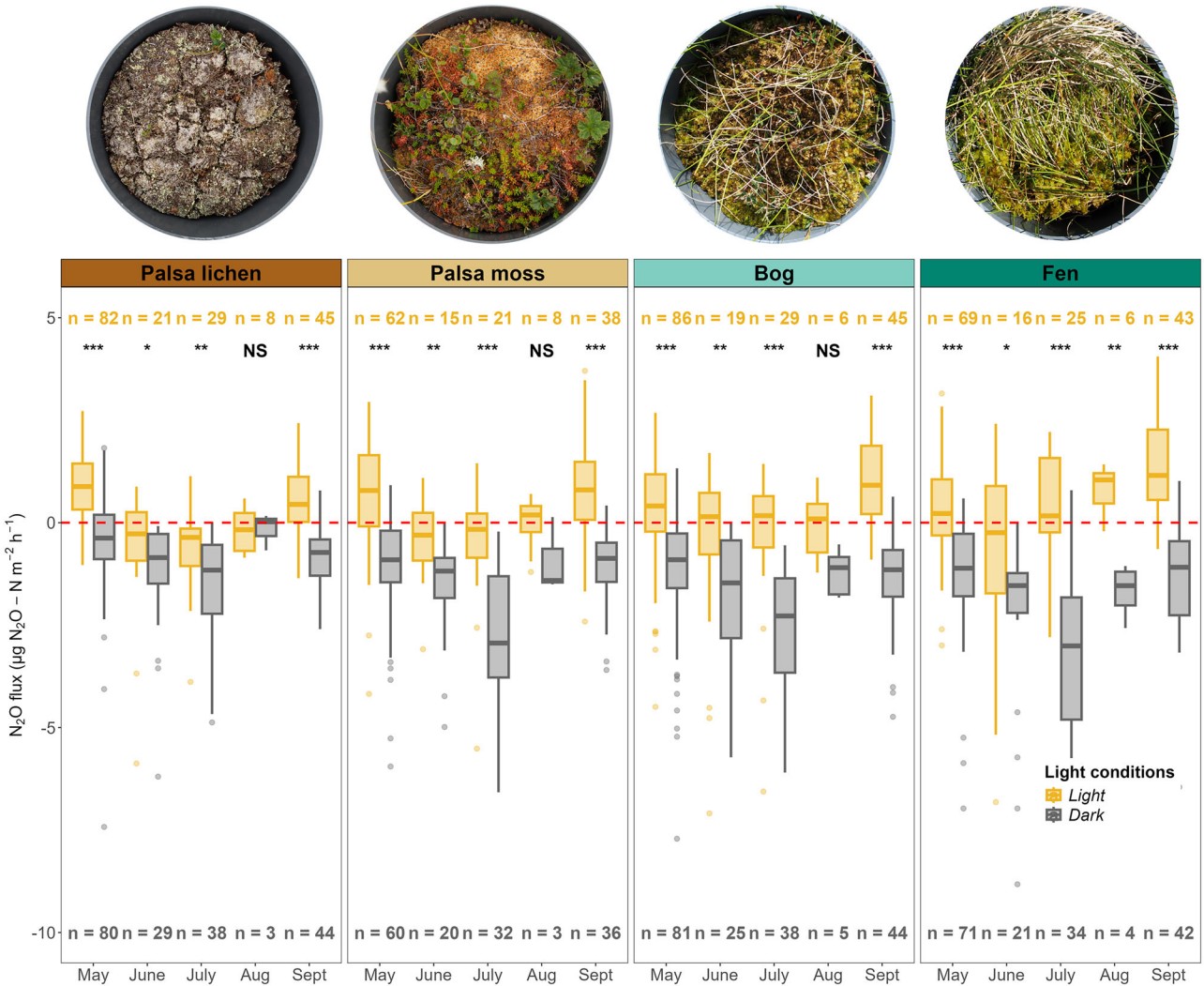

**Fig. 3 | Light (orange boxplots) and dark (gray boxplots) measurements of $N_2O$ fluxes in different micro habitats and months.** *** Designates $p \leq 0.001$, **$p \leq 0.01$, *$p \leq 0.05$ and NS not significant results according to a Wilcoxon signed rank test. The red dashed line marks the border between source (positive flux values) and sink (negative flux values).

$N_2O$ fluxes statistically indistinguishable from zero[29], Li et al. reported significant $N_2O$ uptake in High Arctic tundra soils, and suggested anaerobic denitrification as the driving process[17]. Bhattarai et al. observed net $N_2O$ uptake in laboratory soil mesocosms, whereas field measurements showed net emission[23]. They suggested that using only the top 10 cm of the active layer may have missed deeper $N_2O$ production within the peat profile[30]. Generally, the genetic potential for $N_2O$ reduction is indicated by the presence of the *nosZ* gene, which has been detected in surface layers of uplifted permafrost peatlands[31]. At our site, *nosZ* has been reported mainly in fens but also in the drier palsas[32], implying that anoxic microsites sufficient for complete denitrification may exist even in relatively dry soils. We suggest that several mechanisms can explain net $N_2O$ uptake under aerobic conditions: (i) aerobic denitrifiers can retain *nosZ* activity in the presence of $O_2$[33]; (ii) organic-rich peat creates diffusion-limited microsites that remain anaerobic, allowing local complete denitrification; and (iii) some microbes may assimilate $N_2O$ into biomass via the $N_2$-fixation system[34,35]. Together, the methodological differences and unexplained palsa sink highlight that current permafrost peatland $N_2O$ budgets may still be affected by measurement constraints, including both underestimated uptake due to long chamber closures, the limited amount of $N_2O$ flux data, and incomplete representation of light-driven flux dynamics.

## PAR as contributing driver of $N_2O$ fluxes: higher net $N_2O$ uptake in the dark

Our results show a significant difference between mean light and dark $N_2O$ fluxes across all micro habitats (Wilcoxon rank-sum test 0.37, $p < 0.001$), confirming our second hypothesis. Crucially, light and dark flux measurements were taken within 10–15 min, ensuring minimal change in soil temperature and soil moisture. Under light conditions, fluxes were on average positive, showing net emissions (median (25–75 percentiles): 0.42 (−0.27, 1.17) µg$N_2$O-N m$^{-2}$ h$^{-1}$ ($n = 673$, excluding hot spot)). In contrast, fluxes measured in the dark were consistently negative, indicating net uptake (median (25–75 percentiles): −1.06 (−1.06, −2.05) µg$N_2$O-N m$^{-2}$ h$^{-1}$ ($n = 710$, excluding hot spot)), respectively (Fig. 3). Switching from dark to light increased the flux by around 60%, reversing the sign from net uptake to net emission. Broken up by micro habitat, we found the greatest differences between light and dark fluxes in fen sites, followed by bog, PM, and PL sites, with the largest differences in July, August and September (Fig. 3).

Prompted by our observations in 2022 and 2023, we conducted targeted experiments in 2024 to test whether the differences in light and dark measurements achieved by artificially darkening the chamber during the day-light hours would be confirmed during the night and could thereby exclude measurement artefacts; and whether this phenomenon is specific to

**Fig. 4 | Light–response curves for the $\Delta N_2O$ flux (light flux-dark flux) for each micro habitat, with model fit statistics (pseudo-$R^2$, AIC and RMSE) shown for each curve.** The red dotted line indicates the 0 line. Pseudo-$R^2$ shows the proportion of the variance explained by the model (i.e. goodness of fit of a single model), Akaike Information Criterion (AIC), the model fits (i.e. the relative quality of different models), and root mean square error (RMSE), the accuracy of the model in predicting actual values. Confidence intervals show the 95% bootstrap confidence interval of the mean response curve. The model fit statistics indicate the quality of the model fit for each microhabitat.

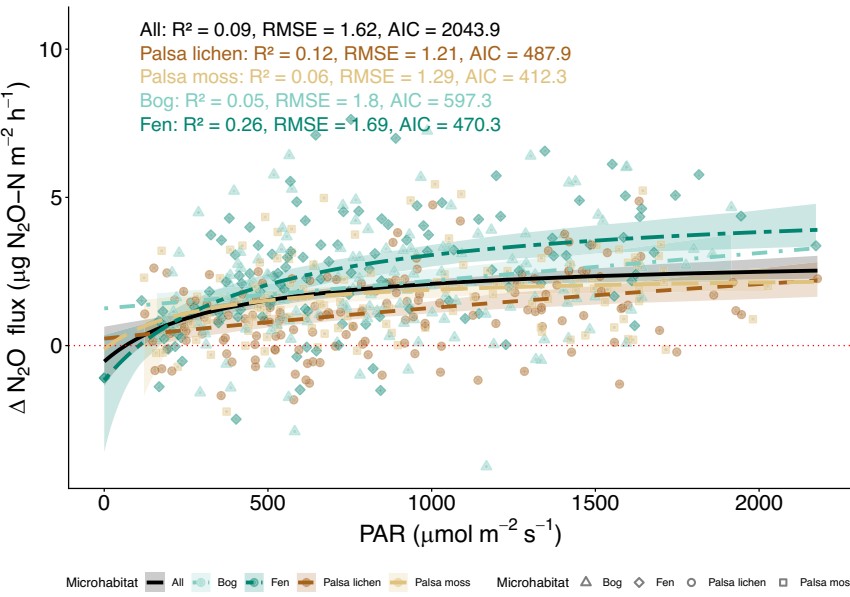

our site, or also visible in other permafrost peatlands. In June, July and August 2024, we thus conducted nighttime measurements ($n = 46$) to check whether $N_2O$ fluxes measured with a transparent chamber in naturally dark conditions during the night would be similar to those measured with an artificially darkened opaque chamber in the daytime. We did these measurements in both polar day conditions in July (i.e. the sun did not set) and dark nights in August. Our measurements confirmed that during nighttime, $N_2O$ was indeed taken up in palsas, bogs and fens, suggesting that the natural day-night cycle may have a more significant impact on $N_2O$ fluxes than can be replicated by artificial covering with opaque chambers (Fig. S5). Further, we collected 24 daytime measurements (4 micro habitats × 6 replicates) in a near-by permafrost peatland in June and July to investigate whether light and dark fluxes would show a similar pattern than at our main study site. This was indeed the case (Fig. S6). We therefore conclude that the light-dark effect we observed is a real phenomenon as reported by previous studies[16,17,23,29], not a measurement artefact, and not exclusively happening at the permafrost peatland of our main study site. This light-dark variability has importance for the diurnal and seasonal variability in net $N_2O$ fluxes in permafrost peatlands, and may sometimes determine whether these systems act as a sink or a source of $N_2O$.

This difference between $N_2O$ fluxes during light and dark conditions implies that the impact of PAR occurs on plants and/or the soil surface; the underlying mechanisms are reacting to changing light conditions within few minutes; and it is most likely a result of multiple interacting drivers. To investigate the strength of the PAR dependency, we used light–response curves, a common tool to model photosynthetic activity, and fitted a rectangular hyperbolic curve to produce a light–response curve for the light-dependent component of the $N_2O$ flux (i.e. $\Delta N_2O$ flux (light flux-dark flux)) according to the Michaelis–Menten equation, which is often used for $CO_2$ studies[36]. Our results suggest that although the typical light–response curves are generally following the trend observed in all micro habitats, it was particularly $N_2O$ fluxes from fens that followed the hyperbolic shape, suggesting that fens show the clearest diurnal cycle for $N_2O$ fluxes driven by changing light conditions ($R^2 = 0.23$, RMSE = 1.83, Fig. 4). While the hyperbolic shape of these curves at the fen site suggests a strong PAR effect, particularly at low PAR values, the large scatter indicates that other important drivers are also at play, particularly in the other micro habitats. According to our data, the differences between $N_2O$ fluxes during light and dark conditions are therefore most likely a result of multiple interacting drivers.

The observed PAR effect can originate from three sources: plants, microbes, or plant–microbe interactions, such as labile C input from plant roots stimulating microbial activity[15]. In our study, the similarity in light–dark flux differences between vegetated palsas (PM) and fens may be explained by higher green canopy cover (GCC) and associated photosynthetic activity during the peak growing season. This is likely due to shrubs on palsas tolerating heat and drought better than *Sphagnum spp.* mosses on bogs, which tend to dry out and turn yellow (Fig. S2). As a result, shrubs and cotton grass on vegetated palsas and fens may exhibit higher light-use efficiency (LUE) under low PAR— consistent with findings in Arctic tundra ecosystems[37–39]. This higher LUE is reflected in similar trends for GCC (Fig. 5) and may enable greater $N_2O$ uptake in dark conditions, when low PAR becomes more limiting (Fig. 3).

Another potential explanation for the light–dark differences in $N_2O$ fluxes is biological $N_2$ fixation, the main N input at our study site[40,41]. The nitrogenase enzyme responsible for $N_2$ fixation is highly light-sensitive and responds rapidly to changes in light conditions[42]; thus, it provides a plausible explanation for the immediate changes in $N_2O$ flux rate when switching from light to dark chamber measurements. Biotic $N_2$ fixation in pristine peatlands is primarily carried out by cyanobacteria, which associate with mosses or live as symbionts in certain lichens[41,43]. Such associations with mosses are widely reported in Arctic ecosystems[40,44,45]; also associations with graminoids have also been reported, although not at our site. Recent studies do, however, confirm the presence of cyanobacteria at our site[46,47], and across the Arctic, approximately 65% of lichens and 44% of bryophytes fix $N_2$[48]. Under light conditions, cyanobacteria convert $N_2$ to $NH_3$, increasing soil $NH_4^+$ pools and potentially stimulating mineral N cycling, including $N_2O$-producing processes[9]. Plant–cyanobacteria interactions may therefore play an important role in regulating N cycling in permafrost peatlands. *Sphagnum* mosses, which dominate anoxic environments such as bogs and fens at our study site, create acidic conditions that favour specific cyanobacterial communities while suppressing many decomposer microbes[43]. In contrast, vascular plants such as sedges and shrubs can modify soil redox conditions through root oxygenation, creating microsites with different N transformation rates[44]. These plant–microbe interactions generate spatial and temporal variability in N availability and may influence both nitrification and denitrification processes responsible for $N_2O$ production[9,48]. In microsites favourable for microbial activity, rapid N turnover may lead to high gross N mineralisation, meaning that nutrients are recycled and delivered more rapidly to plants and microbes[49].

This mechanism may help explain our observations. $N_2$ fixation is strongest in wetter microhabitats such as fens[44,45], where we observed the largest light–dark flux differences. Consistent with this, Basilier et al.

**Fig. 5 | Accumulated local effects (ALE) of photosynthetically active radiation (PAR), net ecosystem exchange/ecosystem respiration, green canopy cover, active layer depth, soil moisture, and soil temperature on N$_2$O fluxes by micro habitat in light (left, yellow background) and dark (right, grey background) conditions.** ALE plots show how predictor variables influence model predictions, on average, while accounting for correlations between them. Mean predictions are centred at 0; positive values show an increase in the predicted N$_2$O flux relative to the mean, negative values indicate a decrease. Ticks along the x-axis indicate the distribution of observed values for each predictor. In parameter ranges covered by many measurements, model estimates are more reliable, whereas sharp changes outside of these regions should be interpreted with caution due to data sparsity.

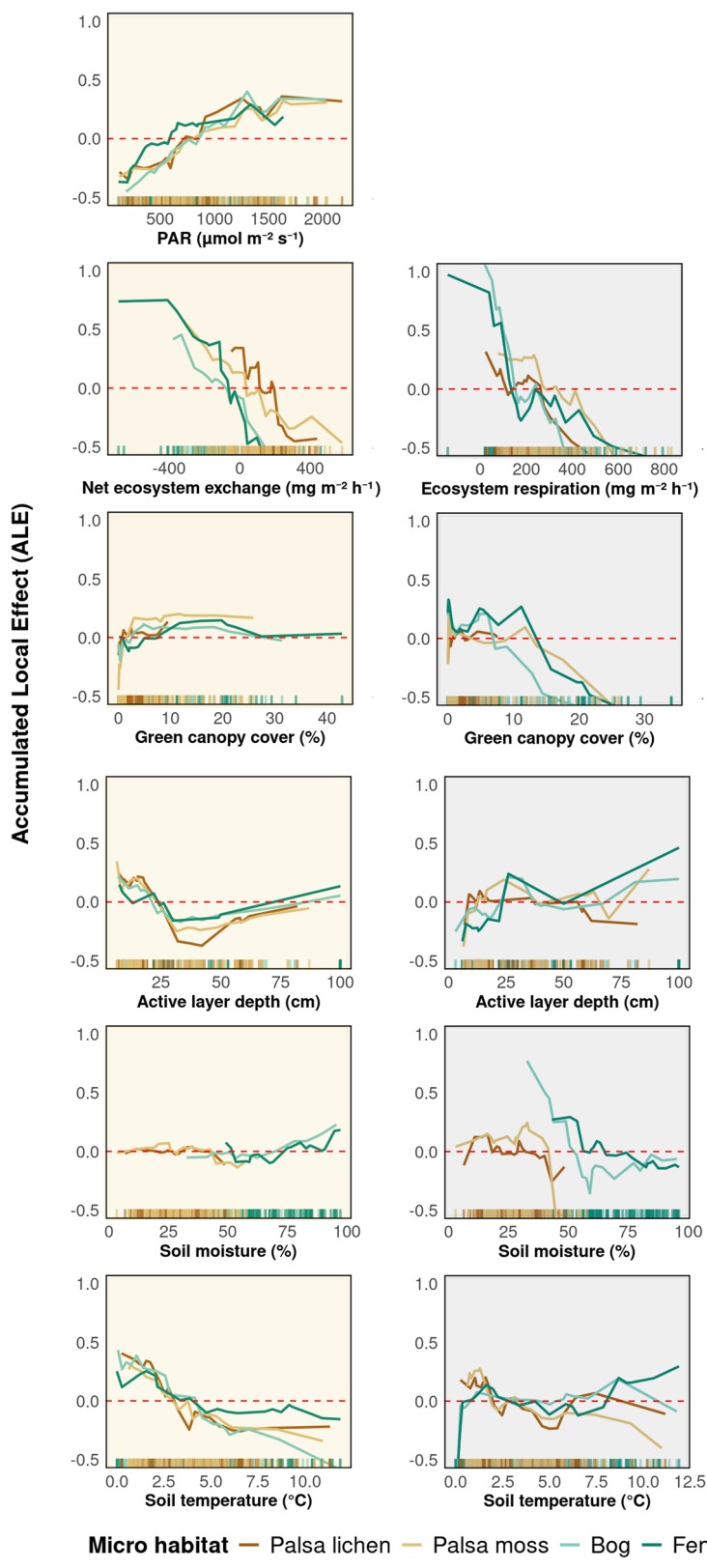

reported a light-dependent linear increase in N fixation at Stordalen with seasonal peaks similar to those in our data[41]. Temperature may further regulate these processes: higher temperatures correlate with increased N$_2$ fixation rates[44,45], with optimal nitrogenase activity around 25 °C[50]. At our site, air and soil temperatures increased during summer (Fig. S2), and air temperatures reached 25 °C in July 2023 and 2024 (Table 1). Differences between microhabitats may also reflect varying cyanobacterial species with distinct sensitivities to temperature and moisture[50].

Future studies should re-evaluate N$_2$ fixation rates at Stordalen, focusing on light responsiveness, species composition across microhabitats, long-term changes over the past 40 years, and the direct role of biological N fixation in N$_2$O dynamics.

**Table 1 | Measurement campaign overview, with season, start date (YYYY-MM-DD), end date, number of observations (N), and min, mean, and max air temperature (°C) during the whole campaign**

| Meas. camp. | Season | Start date | End date | *N* | Min/mean/max air T (°C) |
|---|---|---|---|---|---|
| September 2022 | Late growing | 2022-09-01 | 2022-09-07 | 141 | 6/7.9/9 |
| May 2023 | Early growing | 2023-05-11 | 2023-05-30 | 625 | −1/5.9/14 |
| July 2023 | Peak growing | 2023-07-09 | 2023-07-27 | 192 | 10/16.3/23 |
| September 2023 | Late growing | 2023-09-03 | 2023-09-22 | 210 | 3/10.8/17 |
| June 2024 | Early growing | 2024-06-09 | 2024-06-20 | 143 | 7/13.2/18 |
| July 2024 | Peak growing | 2024-07-17 | 2024-07-26 | 133 | 17/19.6/25 |
| August 2024 | Late growing | 2024-08-20 | 2024-08-21 | 43 | 14/15.4/17 |

**$N_2O$ fluxes driven by interacting soil- and plant-related factors**

Our results highlight that PAR is an important driver of $N_2O$ fluxes, but not the only one, as shown by the large scatter in the light–response of the delta $N_2O$ (Fig. 4). Using accumulated local effect (ALE) plots, we revealed that the variability in the fluxes is driven by multiple parameters, including PAR, $CO_2$ fluxes, GCC, active layer depth (ALD), soil moisture and soil temperature (Fig. 5), with differences in the set of drivers and shape of dependencies under light and dark conditions.

During light conditions, higher PAR, $CO_2$ uptake, as well as shallow active layers and soil temperatures below approx. 3 °C were associated with higher $N_2O$ emissions. Opposite to our expectations, soil temperatures above 3 °C coincided with increased $N_2O$ uptake (Fig. 5); we thus had to reject our third hypothesis. Our results suggest that, opposite to high-emitting, nutrient-rich Arctic permafrost peatlands[3], $N_2O$ emissions at nutrient-poor (sub-) Arctic permafrost peatlands increase at lower temperatures. This correlates with thawing-freezing experiments in boreal soils[51]; indeed, $N_2O$ emissions were higher with lower active layer depth, coinciding with permafrost thaw in spring. We suggest that $N_2O$ may be produced in soils during the snow season and trapped below the ice, and released during permafrost thaw in spring[52]. Even though measurements in permafrost peatlands during the snow-season are needed to confirm this hypothesis, our results strongly hint towards the possibility of $N_2O$ emissions during the colder months, be it from palsas in spring or fens in autumn (Fig. 2). Consequently, incorporating these previously overlooked cold-season fluxes from Arctic permafrost peatlands could markedly increase the estimated terrestrial contribution to the global $N_2O$ budget, underscoring the need to revise climate-model inventories to account for winter-time $N_2O$ emissions and their release during spring thaw.

During light conditions, sunlight triggers photosynthesis, leading to increased $CO_2$ uptake (i.e. negative NEE). This results in more labile C entering the soil through root exudates, which does not only boost heterotrophic activity but also drives denitrification, both directly[53] and indirectly by depleting the soil of $O_2$ through microbial respiration[54,55], particularly when combined with higher soil temperatures. Plants may also release $N_2O$ through their leaves[18], and mediate the transport of microbially produced $N_2O$[18,19]. In a recent study, Karim et al. reported consistent $N_2O$ emissions from the leaves of the pioneer tree species *Salix bebbiana* in temperate and boreal forests[56]. The authors suggested that these emissions may be linked to higher photosynthetic rates that enhance N assimilation in leaves, or to increased stomatal conductance driven by changes in intercellular $CO_2$ concentrations, which could facilitate the release of $N_2O$ from leaf tissues. It is possible that enhanced photosynthesis increased N assimilation and hence $N_2O$ emissions at our site, or that increased stomatal conductance had an effect; however, more research is needed to verify the role of vegetation in emitting or transporting $N_2O$ in Arctic ecosystems.

In contrast, during dark conditions, $N_2O$ uptake increased with $CO_2$ flux (i.e. ER), and soil moisture became a more important driver than soil temperature (indicated by higher ALE values on Fig. 5). The patterns we observe correspond with the conventional thinking of how plant-soil interactions affect microbial N cycling: higher GCC reduces reactive N availability for microbes due to plant uptake, leading to lower $N_2O$ fluxes.

Increasing soil moisture favours complete denitrification, reducing $N_2O$ to $N_2$. One way to explain the increased importance of soil moisture during dark conditions may be related to its association with soil nutrient content. In contrast to our findings for light conditions, where the impact of nutrients appeared negligible (Fig. S7), nutrients other than $NH_{4^+}$, namely Mg, Al, Mn, Zn and Ca, influenced $N_2O$ fluxes during dark measurements (Fig. S8). These nutrients are much higher concentrated in wetter soils (i.e. fens; Fig. S3); palsas are ombrotrophic and receive nutrients only from precipitation and atmospheric deposition, while fens also receive nutrient inputs from surrounding mineral soils. Finally, we saw a tendency toward higher $N_2O$ sources under dark condition with a deeper active layer depth. This might be due to increased soil volume, particularly of the aerobic soil layer, which may favour nitrification and further denitrification, resulting in higher $N_2O$ emissions. This may, at least partially, explain the higher emissions we saw from fens overall, as well as the higher emissions from all micro habitats in September (Figs. 3 and 5).

Overall, our study revealed a shift from temperature-dominated fluxes during light to moisture-dominated fluxes during dark conditions, at which $N_2O$ emissions transition to sinks. We also found that soil moisture and ALD are closely linked, with their impacts on $N_2O$ fluxes varying significantly between dark and light conditions. Notably, our study also highlights the interdependent relationship between $CO_2$ and $N_2O$ fluxes, with higher $CO_2$ uptake leading to higher $N_2O$ emissions under light conditions and $CO_2$ emissions stimulating higher $N_2O$ uptake under dark conditions.

**$N_2O$ hot spot on vegetated palsa**

Our measurements highlight the importance of a sufficient amount of replicates in heterogeneous ecosystems like the Arctic: in the otherwise $N_2O$-consuming peatland, one of our chamber base positions, a vegetated and N-limited PM, acted as a continuous hot spot over the three years of our study (Fig. 6, see hot spot boxplot, $n = 79$). To the best of our knowledge, no Arctic study has reported a vegetated hot spot before, while the hot spot activity of bare peat patches on permafrost peatlands has been studied in detail[3,4,22]. When we include this hot spot, our study site may act as a net $N_2O$ source, with median (25–75 percentiles) $N_2O$ fluxes of −0.28 (−1.24, 0.68) $\mu gN_2O$-N m$^{-2}$ h$^{-1}$ ($n = 1462$), and PM showing low emissions of median (25–75 percentiles) 0.01 (−1.03, 1.53) $\mu gN_2O$-N m$^{-2}$ h$^{-1}$ ($n = 344$). Motivated by this hot spot, we extended our search of similar high-emitting locations across the site, including bare peat surfaces where hot spots have previously been observed[3]. Only one further hot spot was detected (Fig. 6, see Bare boxplot on the very left, $n = 2$), for which the soil $NH_{4^+}$ content was much higher than for the other chamber base positions (data not shown). The presence of hot spots could significantly enhance the landscape-integrated $N_2O$ emissions due to the disproportionally high flux, and maybe even change the site from a net sink to a net source.

The $N_2O$ emissions in the vegetated hot spot showed a similar light-dependency to the other chamber base positions featuring low fluxes, with higher $N_2O$ emissions in the light, but instead of $N_2O$ uptake, lower $N_2O$ emissions in the dark (Fig. 7a). Fluxes showed a relatively clear seasonal pattern with highest emissions in the peak summer, which is in line with

**Fig. 6 | N₂O fluxes in the Stordalen mire across three snow-free seasons, including a bare hot spot (bare) found in specific hot spot screening and the continuous vegetated hot spot (hot spot), with PL and PM indicating palsa lichen and palsa moss, respectively.** Please note that for better visualisation, 'bare' is shown as boxplot despite low sample size (*n* = 2). 'Hot spot' only contains the emissions from the continuous hot spot. The red dashed line marks the border between source (positive flux values) and sink (negative flux values). Note that all N₂O fluxes measured in both light and dark conditions are shown.

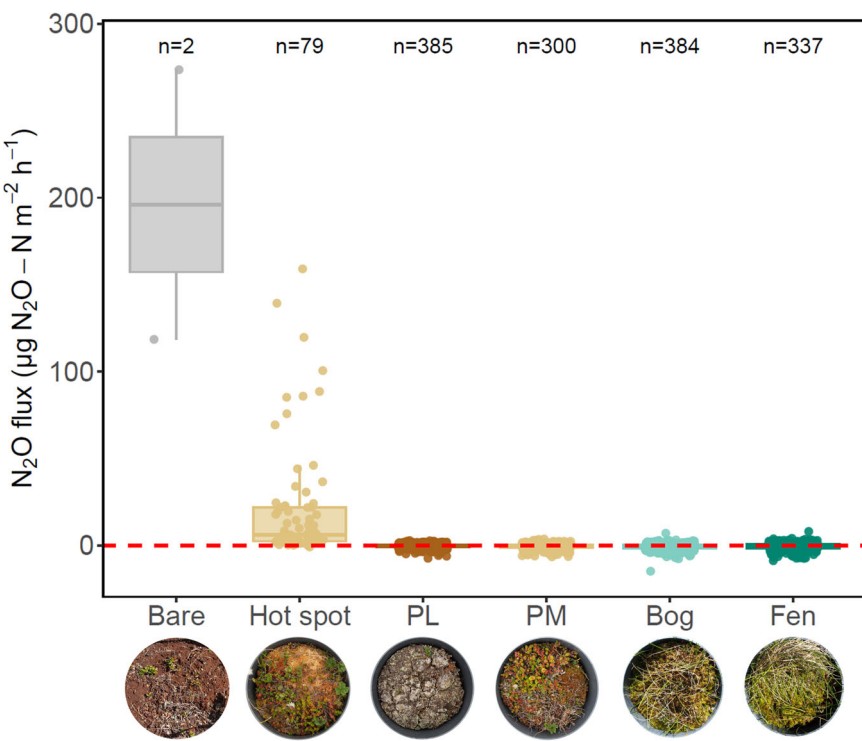

previous studies showing that higher temperature leads to higher N₂O emissions when moisture or N availability are not limiting N₂O production[3]. In September 2022, our one-week campaign coincided with warm and dry conditions, and N₂O emissions from the hot spot were high (Fig. 7a). In contrary, in September 2023, a colder and wetter autumn, N₂O emissions remained low. The hot spot was mostly driven by ER ($R^2$ = 0.63; Fig. 7f) and NEE ($R^2$ = 0.59; Fig. 7f), followed by GCC ($R^2$ = 0.42 and 0.49 for dark and light measurements, respectively; Fig. 7d), soil temperature ($R^2$ = 0.38; Fig. 7b), and soil moisture ($R^2$ = 0.23 and 0.32 for dark and light measurements, respectively; Fig. 7c). ALD (Fig. 7e) only played a minor role. However, when we exclude the influence of other predictors, only GCC and $CO_2$ flux remain positively correlated with N₂O (Kendall's $\tau$ 0.32 and 0.28, respectively). Overall, the hot spot showed a positive correlation with all environmental variables except soil moisture (Fig. 7).

The positive correlation between N₂O and both NEE fluxes and ER emissions of the hot spot aligns with most studies on $CO_2$-N₂O relationships[20,57]. In an agricultural field in the UK, Alskaf et al. found a positive relationship between NEE and N₂O emissions and hypothesised that these processes are governed both below- and aboveground, i.e. through soil or plant emissions[58]. When, and shortly after, plants photosynthesise, they release labile organic compounds into the soil through their roots, which is expected to increase both $NO_3^-$ and $NH_{4^+-N}$ in the soil[59]. Since N₂O is produced from nitrate during denitrification and ammonium during nitrification[60], this leads to an increased N₂O production[59]. On our site, $NO_3^-$ concentrations in the surface peat were below the detection limit of the method in all micro habitats, and $NH_4^+$ concentrations at the hot spot were not higher than on other PM sites; however, we cannot rule out the possibility of elevated mineral N content in the deeper layers underneath the hot spots.

The vegetation composition and GCC on the hot spot was similar to other PM replicates, and did not differ in soil moisture, soil temperature, nutrients, or any other measured environmental variable. When we did some test measurements around the hot spot, we found that the surrounding area also showed elevated fluxes, particularly along a distinct, but barely visible line on the ground surface—possibly an early-stage crack in the permafrost peatland (data not shown). While the exact cause of this hot spot remains unclear, its occurrence may be related to this linear feature.

One possibility is that N₂O is released from deeper layers within the permafrost, which are known to be more mineral N-rich at our study site[61]. Since we observe the highest N₂O flux rates during the peak growing season (July), optimal soil conditions in terms of elevated soil temperatures and higher thaw depth may drive these peaks. However, the source of our N₂O hot spot may not be in the bottom part of the active layer, but still deeper than the surface peat where we did the nutrient measurements. In those intermediate layers, favourable conditions for N₂O production, including low C/N ratio (<25) and favourable intermediate soil moisture content for coupled nitrification-denitrification, might prevail[30].

Overall, the presence of this hot spot on a vegetated palsa remains puzzling and needs further investigation, e.g. through microbial analyses (is the microbial community different?), N₂O measurements at different depths (where is the source of the emitted N₂O?), or methods that can look into the soil column and provide insights on ground-ice features and permafrost characteristics (is there a relationship between the hot spot and permafrost characteristics?), such as ground-penetrating radar. The presence of a hot spot might be concerning for the N₂O upscaling studies that are conducted without accurately capturing the small-scale heterogeneity of N₂O fluxes over these ecosystems, and may indicate potential for increased emissions with continued warming. Furthermore, hot spots may also significantly bias eddy covariance measurements that capture these hot spots without knowing of them, and thus assigning inaccurate average flux rate to the larger area. Drone-based methods might be useful to localise these hot spots, as they can conduct large-scale observations while capturing small-scale features[62]. Overall, our findings underscore the complex, microhabitat-specific controls of N₂O in Arctic permafrost peatlands and provide critical data for improving upscaling and modelling of greenhouse gas fluxes under future climate change scenarios.

In conclusion, our results demonstrated that most micro habitats in the (sub-) Arctic permafrost peatland consistently functioned as N₂O sink during the growing season, with a micro habitat-specific variability: dry palsas exhibited the weakest sink, wetter bog and fen the strongest. Seasonal dynamics were pronounced, with emissions from PL in May and fen in September, and peak sink strength during the warmest summer months. We found significant differences between light and dark N₂O fluxes across all micro habitats in all months (except August due to insufficient data),

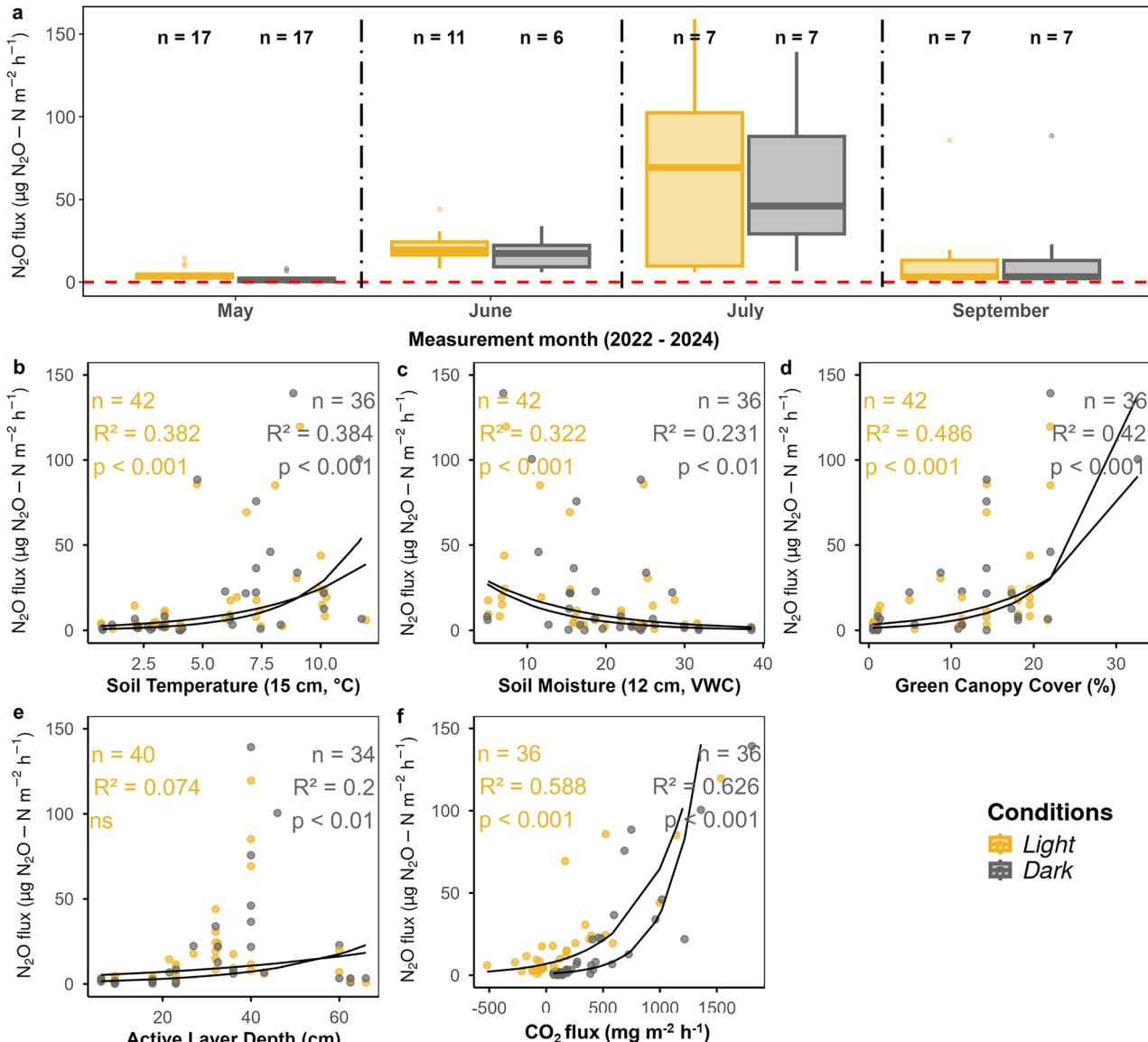

**Fig. 7 | Environmental drivers of N2O emissions on a single vegetated palsa hot spot.** Seasonal differences between the measurement months (2022-2024, excluding interannual variability) with orange boxplots showing light, and grey boxplots showing dark measurements, respectively (**a**). The red dashed line indicates the border between source (positive values) and sink (negative values). Other plots show simple regression models between the $N_2O$ flux and **b** soil temperature at 15 cm depth, **c** volumetric water content (%), i.e. soil moisture at 12 cm depth, **d** green canopy cover, **e** active layer depth, and **f** $CO_2$ flux. We tested linear, quadratic, logarithmic and exponential models and compared them by AIC. The lowest AIC for all predictors were given by the exponential model: $y = e^{\beta_0 + \beta_1 x}$, where $y$ is $N_2O$ flux and $x$ is the predictor.

which suggested a diurnal and seasonal cycle behaviour in nutrient-poor Arctic permafrost peatlands. This also highlighted the need to include light $N_2O$ measurements in future chamber-based studies in the Arctic to re-asses the circumpolar $N_2O$ budget[63]. Across all micro habitats, photosynthetically active radiation (PAR) and ER / NEE were key drivers, which may be partly explained by their association with the combined effect of soil temperature, soil moisture and ALD. To the best of our knowledge, our $N_2O$ dataset is, to date, the most extensive Arctic $N_2O$ dataset. Nevertheless, our measurements were not continuous over 1 year, which introduces uncertainties since $N_2O$ fluxes are temporally very variable. We suggest that future Arctic GHG flux research includes $N_2O$ measurements from both transparent and opaque chambers, ideally from both manual and automated chamber combined to adequately address both spatial and temporal variability of fluxes. We further emphasise the need to investigate the light-dependency of $N_2O$ fluxes in more detail; microbial studies, measurements

on plant $N_2O$ fluxes, and incubation studies with controlled laboratory conditions will be essential for advancing our understanding.

## Methods

### Study site

The majority of data were collected during daytime in the Stordalen mire, a complex palsa mire underlain by sporadic permafrost located in subarctic Sweden (68° 20.0′ N, 19° 30.0′ E), 10 km east of Abisko (Ábeskovvu in Northern Sámi language). Permafrost has been rapidly thawing in this region over the last decades, and only remains in the dry uplifted areas on the peatland (palsas)[64]. Vegetation on the palsa is mainly dominated by lichen (*Cladonia spp.*), shrubs (*Empetrum hermaphroditum, Betula nana, Vaccinium uliginosum, V. vitis-idaea* and *Rubus chamaemorus*) and some mosses (*Dicranum elongatum, Sphagnum fuscum*). Both bogs and fens contain peat-forming mosses (*Sphagnum balticum, S. lindbergii* and *S. riparium*), with the

**Table 2 | Details of the $Q_{10}$ functions fitted to the ER and chamber temperature**

| Micro habitats | $R^2$ | $Q_{10}$ functions |
|---|---|---|
| Palsa lichen | 0.31 | $1.18 \times 1.67^{(T_{chamber}-10)/10}$ |
| Palsa moss | 0.41 | $1.69 \times 1.96^{(T_{chamber}-10)/10}$ |
| Fen | 0.41 | $1.35 \times 2.23^{(T_{chamber}-10)/10}$ |
| Bog | 0.57 | $1.21 \times 2.10^{(T_{chamber}-10)/10}$ |

dominant vascular plants in fens being cotton grass (*Eriophorum vaginatum*, *E. angustifolium*) and in bogs sedges (*Carex rotundata*, *C. rostrata*). For our study, we randomly selected 24 chamber base positions (measurement plots) on a dry-to-wet thawing gradient from palsa to bog to fen, with 6 replicates for each land cover type: palsa lichen, palsa moss, bog and fen (see Table S1 for coordinates and number of replicates, and[26] for more information).

To investigate if the differences in $N_2O$ fluxes measured in light and dark conditions during the day were an artefact, we conducted additional nighttime measurements at Stordalen (19/20 June, 17/18 July, 20/21 August 2024 between 20:00 and 02:00 Swedish local time, UTC + 2; $n = 46$). We also did some 'hot spot screening' on vegetated palsas and bare soils. To test our hypothesis that $N_2O$ fluxes were higher in light compared to dark conditions, we further collected data at the Storflakket mire (68.347209, 18.971356) near Stordalen during 2 days in June and July 2024.

### Flux sampling

Data were collected in September 2022, May, June, July and September 2023, and June, July and August 2024 (Table 1). For our measurements, we used a custom-built static, non-steady state, non-flow-through chamber[65] made from acrylic glass (Göli GmbH, Germany, light transmittance of 92–93%) with a height of 250 mm and a diameter of 250 mm, equipped with a fan (SUNON Maglev, 80 mm × 80 mm × 25 mm, 2000 RPM) to ensure well-mixed conditions within the chamber during the measurements. Additional sensors included a relative humidity and temperature probe (EE08, E+E Elektronik, Germany) and a pressure sensor (61402V, RM Young). As complementary variables, we measured soil temperature at 15 cm depth (PT100 4-wire sensors, JUMO GmbH & Co. KG) at each quadrant outside of the plot, soil moisture at 12 and 30 cm (CS655-DS and CS650-DS, Campbell Scientific), and photosynthetically active radiation (PAR) (PQS1, Kipp and Zonen). To measure $N_2O$ and $CO_2$ concentrations, we used the Aeris MIRA Ultra $N_2O/CO_2$ analyser (Aeris Technologies; sensitivity: 0.2 ppb/s for $CO_2$ and $N_2O$, frequency: 1 Hz). For the dark measurements, a custom-made, reflective, light-impermeable tarp was placed on top of the transparent chamber. In the field measurements, the chamber closure time for each measurement was 300 s for all GHG, based on ref. 26. In 2024, we added plant root simulator (PRS) probes (Western AG Innovations Inc., Canada) next to our chamber base positions to get information on the nutrient status at the different micro habitats.

All data were processed in R (version 4.5.1)[66] and version controlled in GitLab (https://git.bgc-jena.mpg.de/ipas/fluxprogeniereleases.) A filter script was applied to pre-process and quality-control the raw data, such as removing data points within a specific time interval at the start of the measurement period to account for the time lag of gases moving through the tubes to reach the laser cell. The filter script also included quality control of other parameters by, e.g. removing implausible values (e.g. −9999), replacing negative PAR values with 0, averaging soil temperature gained from the four sensors, and setting minimum and maximum values for all parameters. From our original 1654 measurement periods, 167 (10.1%) were removed through this quality control. More information on the chamber measurement setup, instruments and the script can be found in ref. 26.

### Flux calculations

In this study, negative flux values indicate sinks (i.e. uptake by the ecosystem) and positive values sources (i.e. emissions to the atmosphere) of greenhouse gases. To calculate the GHG fluxes, we used the R goFlux package[67]. For both $N_2O$ and $CO_2$, we used the non-linear HM model[68,69] with all 300 and 210 data points (seconds), respectively. For $N_2O$, we used the best.flux function provided by goFlux to restrict the curvature of the model (g.factor = 4)[67,69]. For $CO_2$, we cut the original chamber closure time of 300 s down to 210 s, which was still enough to promote a non-linear curvature as suggested by ref. 70. We then exclusively used the non-linear fluxes, which we further corrected with a temperature-response curve. To obtain GPP and ER fluxes from chamber measurements, we first fit four $Q_{10}$ curves given in Table 2, per each microhabitat, between chamber temperature and $CO_2$ fluxes measured under non-transparent chamber condition (i.e. ER). Subsequently, we used these relationships to estimate the ER fluxes for transparent chamber measurements, which then can be used to estimate the GPP fluxes. Here, we assume that the relationships between ER and temperature will stay unchanged under different levels of PAR. The details of these fittings are given in Table 2.

### Statistical analyses

To explore the drivers of the observed $N_2O$ fluxes, we applied random forest (RF) models to the entire dataset, including all observations rather than to subsets, in order to identify the most important predictors (see below). This initial screening indicated that variables related to light and vegetation activity ($CO_2$ fluxes, PAR and GCC) were among the most important factors. To assess differences in $N_2O$ fluxes between light and dark conditions, we used a Wilcoxon rank-sum test (Mann–Whitney $U$ test), a non-parametric alternative to the t-test that does not assume normality of the data. Motivated by the significant results, we divided the dataset in three complementary ways:

1. Light measurements – to assess $N_2O$ fluxes under light conditions.
2. Dark measurements – to assess $N_2O$ fluxes without direct light influence (noting that, unlike $CO_2$ uptake, $N_2O$ processes can also occur in the dark).
3. Light-dependent component – defined as the difference between light and dark $N_2O$ fluxes, to isolate the effect of light.

To test for differences in $N_2O$ fluxes among micro habitats (PL, PM, bog and fen), we performed a one-way analysis of variance (ANOVA) with habitat type as the predictor and a Tukey's honestly significant difference (HSD) test for post-hoc pairwise comparisons to identify which habitats differed significantly. We then reported mean and median $N_2O$ flux values. Because we found statistically significant differences between the $N_2O$ fluxes of the different micro habitats, except between fen and bog, we further divided our results in micro habitat-specific drivers. Divided into these subsets (i.e. light/dark, light-dark, plus micro habitat), we analysed the main drivers over all micro habitats and years. Unless otherwise indicated, we excluded one hot spot from all statistical analyses and analysed this hot spot separately.

All statistical analyses were performed in R version 4.5.1[66], using the packages indicated below. Because most of our data were not linearly distributed and environmental drivers correlated to each other, we used various statistical approaches, namely (a) RF models, (b) principal component analysis (PCA) paired with RF models, (c) RF-based accumulated local effects (ALE) analyses, (d) non-linear regression models for the light–dark fluxes, and (e) exponential regression models for the $N_2O$ hot spot. The first RF models were built to investigate the overall impact of all environmental drivers on $N_2O$ fluxes (a). The hyper-parameters of the random forest models were optimised using a grid search technique coupled with cross validation (R *mlr3* package). With these optimised parameters, 100 different RF models were built, each trained and validated on different training- and test-data splits. For the final feature importance, we used the best 10 models with highest $R^2$ scores (Figs. S7 and S8). With these models, we were also able to estimate the impact of the nutrients measured at our location (b). Since the Stordalen mire is nutrient-poor, none of the nutrients were statistically significantly related to $N_2O$ fluxes (partial correlation $p > 0.05$, see SI). Nitrate ($NO_{3-}$), copper (Cu) and cadmium (Cd)

concentrations were below the PRS probes' detection limit. To examine the importance of nutrients for $N_2O$ fluxes at our site, we first used PCA to reduce the dimensionality and used the components that explained most of the variations in the RF model. Three quarters of the nutrient effect on the variability could be mostly explained by manganese (Mg), aluminium (Al), sulfur (S), calcium (Ca), potassium (K) and $NO_{3-}$ (principle component analysis (PCA): PCA 1, proportion of variance: 0.49, contributors Mg and Al 0.41; PCA 2, proportion of variance: 0.16, contributors S and Ca 0.53; PCA 3 proportion of variance: 0.10, contributors K and $NH_{4+}$ 0.74 and 0.41, respectively). When we included nutrients in our analyses (PCA 1–3 in RF on light fluxes of all micro habitats), we found that nutrients did not explain the variability of $N_2O$ fluxes, whereas air temperature and $CO_2$ fluxes (NEE) explained more than 20% each (RF $R^2$ = 0.723, mtry = 3, max depth = 30, ntrees = 500, Fig. S7). However, $NH_{4+}$ values were generally higher in palsa than bog and fen, and significantly correlated to PM sites (Fig. S4). In contrast to our findings for light conditions, nutrients other than $NH_{4+}$, namely Mg, Al, Mn, Zn and Ca, influenced $N_2O$ fluxes during dark measurements (RF $R^2$ = 0.79, mtry = 6, max depth = 37, ntrees = 1500, Fig. S8).

To quantify the marginal influence of environmental predictors on $N_2O$ fluxes across micro habitats, we implemented a RF-based ALE analysis (c). For this, we excluded outliers within each micro habitat using the interquartile range (IQR) method. We used green canopy cover (GCC), PAR, $CO_2$ flux (NEE for light, ER for dark), soil moisture at 12 cm, soil temperature at 15 cm, and active layer depth as predictors. After fitting a RF model using the entire dataset (R *randomForest* where, $R^2_{light}$ = 0.38, $RMSE_{light}$ = 0.73 $\mu gN_2O\text{-}N\,m^{-2}\,h^{-1}$ and $R^2_{dark}$ = 0.28 and $RMSE_{dark}$ = 0.91 $\mu gN_2O\text{-}N\,m^{-2}\,h^{-1}$), we generated ALE plots for each predictor (R *iml*) divided into specific microhabitat to quantify the average marginal effect of each predictor on $N_2O$ fluxes while accounting for non-linear relationships and interactions between predictors. This is done by estimating the first-order effects by computing local differences in model predictions along the distribution of each predictor, and integrating these across all observations. The resulting ALE plots were compared across micro habitats to identify habitat-specific sensitivities of $N_2O$ flux to environmental drivers.

To evaluate light–dark differences in $N_2O$ fluxes, we constructed light–response curves by fitting non-linear regression models of the form

$$\Delta N_2O\,\text{flux} = \frac{a \cdot \text{PAR}}{b + \text{PAR}} + c$$

where $a$ represents the asymptotic maximum response, $b$ the half-saturation constant, and $c$ a vertical offset (d). Prior to model fitting, we excluded anomalous values and restricted the dataset to valid PAR and flux measurements. Models were fit separately for each microhabitat (PL, PM, bog, fen) as well as for the full dataset. Confidence intervals were estimated using a non-parametric bootstrap (200 resamples) by refitting the model to resampled datasets and extracting the 2.5% and 97.5% quantiles of predicted values at each PAR level. Model fitting was carried out using non-linear least squares with parameter bounds to ensure biologically meaningful estimates. For each fitted model, we extracted parameter estimates ($a$, $b$, $c$) with associated $p$-values, as well as model fit statistics including pseudo-$R^2$, Akaike Information Criterion (AIC), and root mean square error (RMSE). These metrics were used to compare the performance of light–response models across habitats. To systematically assess how the $N_2O$ hot spot responded to different environmental drivers, we fitted separate exponential regression models for each predictor variable (e). For each predictor, fluxes were log-transformed and regressed against the variable of interest. Model performance was summarised by sample size, the coefficient of determination ($R^2$), and the significance of the predictor term ($p$-value).

## Data availability

The experimental data that support the findings of this study are available in Edmond with the identifier: https://doi.org/10.17617/3.WOIQRC.

## Code availability

The scripts for processing and analysing the data are publicly available at https://git.bgc-jena.mpg.de/ipas/fluxprogeniereleases under the terms of the GNU General Public License version 3.

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

## Acknowledgements
The authors thank the service groups at the Max Planck Institute for Biogeochemistry for their help before and during the field campaign. We also thank Antonin Affolder, Mattias Dalkvist, Valentin Kriegel, McKenzie Kuhn, Alena Markelova, Bailey Mullins, Mark Schlutow, and the staff from the Abisko Scientific Research Station, particularly Erik Lundin and Niklas Rakos, for their help in the field. Many thanks to Jan Engel for his help with the FluxProGenie script and Annett Boerner for the graphical abstract.

## Author contributions
N.Y.T., M.G., C.B. and M.E.M. designed the study. N.Y.T., M.R., R.E.L. and W.H. gathered the data in the field. N.Y.T., A.B. did the statistical analyses, supported by K.I., T.Y., N.J.E., A.M.V., M.E.M., D.P., T.V. and M.G. for both statistics and interpretation. N.Y.T. wrote the first version of the manuscript with the help of A.B., after which all co-authors provided input on the manuscript text, figures and discussion of scientific content.

## Funding
N.Y.T., A.B., K.I., T.Y., N.J.E. and M.G. disclose support for the research of this work from the European Research Council (ERC) under the European Union's Horizon 2020 research and innovation programme (grant agreement No 951288, Q-Arctic). T.V. discloses support for publication of this work from ICOS-Finland (University of Helsinki). M.E.M., W.H. and D.P. disclose support for publication of this work from the Research Council of Finland-funded Thaw-N project (no. 349503, no. 353858). A.-M.V. discloses support for publication of this work from the TED Audacious Project (Permafrost Pathways). C.B. discloses support for publication of this work from the Austrian Science Fund (project PERNO, no. 10.55776/M3335) and the Research Council of Finland through project N-PERM (no. 341348). Open Access funding enabled and organized by Projekt DEAL.

## Competing interests
The authors declare no competing interests.

## Additional information
**Supplementary information** The online version contains Supplementary material available at https://doi.org/10.1038/s43247-026-03698-3.

