## [Transparent Peer Review File · Communications Earth & Environment]

Light and dark conditions control the nitrous oxide uptake and emission dynamics in a subarctic, nutrient-poor permafrost peatland

Corresponding Author: Dr Nathalie Triches

Version 0:

Decision Letter:

Dear Mx Triches,

Your manuscript titled "Between light and dark, source and sink: N₂O dynamics in a subarctic, nutrient-poor permafrost peatland" has now been seen by 3 reviewers, whose comments are appended below. You will see that they find your work of some potential interest. However, they have raised substantial concerns that must be addressed. In light of these comments, extensive revisions will be required before we can further consider the manuscript for publication. We would, however, be interested in considering a revised version that fully addresses these serious concerns.

We hope you will find the reviewers' comments useful as you decide how to proceed. Additionally, the following points should be addressed in the revised manuscript:

- *Thoroughly reorganize the paper structure with a clear experiment conceptual figure and clear hypotheses;
- *Add more detailed analyses to distinguish light-effect from other factors or focus on light-induced effect not direct effect of lights;
- *Add some data about microbial characteristics, essential plant traits, and soil organic carbon or strength their discussion for inferred mechanisms.

If additional work allows you to either incorporate or refute these criticisms, we will be happy to look at a substantially revised manuscript. If you choose to take up this option, please either highlight all changes in the manuscript text file, or provide a list of the changes to the manuscript with your responses to the reviewers.

When resubmitting, please provide a point-by-point response to the reviewers' comments. Please submit your responses as a separate file, distinct from your cover letter where you can add responses to the Editors' comments that you do not want to be made available to the reviewers. Word files are preferred. We recommend that any figures, tables or graphs that are included in the response to reviewers are also included in the main article or Supplementary Information.

If the revision process takes significantly longer than three months, we will be happy to reconsider your paper at a later date, as long as nothing similar has been accepted for publication at Communications Earth & Environment or published elsewhere in the meantime.

Please use the following link to submit your revised manuscript, point-by-point response to the reviewers' comments with a list of your changes to the manuscript text (which should be in a separate document to any cover letter), a tracked-changes version of the manuscript (as a PDF file) and any completed checklist:

Link Redacted

Please do not hesitate to contact us if you have any questions or would like to discuss the required revisions further. Thank you for the opportunity to review your work.

Best regards,

Huai Chen, PhD
Editorial Board Member
Communications Earth & Environment
orcid.org/0000-0001-7650-289X

Somaparna Ghosh, PhD
Associate Editor - Communications Earth & Environment
Consulting Editor - Communications Sustainability

EDITORIAL POLICIES AND FORMAT

If you decide to resubmit your paper, please ensure that your manuscript complies with our editorial policies and complete and upload the checklist below as a Related Manuscript file type with the revised article:

- Behavioural and social science
- Ecological, evolutionary & environmental sciences
- Life sciences

For your information, you can find some guidance regarding format requirements summarized on the following checklist: (<https://www.nature.com/documents/commsj-phys-style-formatting-checklist-article.pdf>) and formatting guide (<https://www.nature.com/documents/commsj-phys-style-formatting-guide-accept.pdf>).

REVIEWER COMMENTS:

Reviewer #1 (Remarks to the Author):

I commend the author for the valuable work conducted in these regions and for highlighting the decisive role of light conditions in determining flux direction (uptake during dark periods and emission during light periods). Given the scarcity of N₂O data in these areas, this finding offers important insights for the Arctic greenhouse gas budget. Especially as the authors mentioned on Lines 136-138 'To the best of our knowledge, ours is the first study to report a persistent, albeit small, ecologically relevant N₂O sink in a subarctic permafrost peatland over several years, and the first to observe continuous net uptake on the dry, uplifted tundra surfaces'. The research content is suitable for publication in Communications Earth & Environment. Although the large sample size from three years of greenhouse gas observations is statistically compelling, there is still a lack of in-depth mechanistic investigation in the current study. The present version has certain limitations. I have provided some comments below, which I hope the author will find useful when revising the manuscript.

Comments to Authors

1. In the Abstract section, the author directly emphasizes light as the driving force of N₂O flux. I have some doubts about the credibility of this claim. I do not deny that light may have an impact on N₂O flux, but this impact is likely not due to light directly affecting the N₂O production process. Instead, it may indirectly affect N₂O flux by influencing plant physiological processes, or it may also be due to temperature differences. However, this article omits the role of plants and temperature, directly expressing the impact of light on N₂O flux, which seems inappropriate.

2. As mentioned in the third paragraph of the introduction, there remains an urgent need for long-term, high-resolution field observations. This study presents a three-year observation period with a substantial sample size, which is statistically robust. However, it lacks more in-depth experimental investigations into the mechanisms of N₂O, such as functional gene analysis related to N₂O production, isotopic characterization of N₂O, or tracer studies. To address these gaps while highlighting the strength of the large sample size, I recommend that the authors supplement their work by conducting a meta-analysis that aggregates global literature data from similar research areas (e.g., northern permafrost peatlands). Although collecting such data may require additional time, it would significantly enhance the global extrapolation and comparability of the study's findings.

3. We have observed that N₂O fluxes are relatively low in terrain dominated by lichens and mosses (PL/PM). Considering the close association between mosses (and to some extent lichens) and nitrogen-fixing cyanobacteria, I suggest that the discussion could further explore the impact of interactions between different functional plant types and soil microorganisms on nitrogen cycling in peatlands. Additionally, within the denitrification pathway, functional genes such as nirK (alongside others like nirS) are also involved in N₂O production and consumption. Is there any quantitative data or research on these functional genes to support this point, given that this study did not include measurements of soil microbial indicators?

4. I would like to highlight that different functional plant types in peatlands may have significant impacts on greenhouse gas emissions. For example, some herbaceous plants in permafrost peatlands can access more available nitrogen through their deeper root systems. These plants may directly emit greenhouse gases by bypassing oxidation in the aerobic layer. Therefore, reporting the coverage and abundance of such plants at each observation site could help us better understand the dynamics of N₂O emissions.

5. Lines 276-279, Plant leaf-driven N₂O fluxes primarily result from either the transport of soil-produced N₂O via photosynthesis/transpiration or are associated with phyllosphere microorganisms. Some progress has been made in understanding these processes in woody plants, which may be useful for the discussion here:

Reference :

Foliar N₂O emissions constitute a significant source to atmosphere.

Foliar methane and nitrous oxide fluxes in *Salix bebbiana* respond to light and soil factors.

6. Lines 433-434, Could the authors please introduce the light transmittance of acrylic glass?

7. Lines 441-444, 'To measure N₂O and CO₂ concentrations, we used the AERIS MIRA Ultra N₂O/CO₂ analyser (AERIS Technologies; sensitivity: 0.2 ppb/s for CO₂ and N₂O, frequency: 1 Hz); to measure CO₂ and CH₄, we used the Li-7810 CH₄/CO₂/H₂O Trace Gas Analyser from LI-COR'. I did not see the CH₄ results in this manuscript.

8. It is recommended that the proportion of invalid values removed be reported, as I would like to know how many observations were excluded.

9. In Figure S1, "n=?" seems unclear. Was the sample size not reported here?

10. In Figures S6, I did not see the PAR. So what is the importance of PAR to N₂O flux?

Reviewer #2 (Remarks to the Author):

This study is interesting and shows the N₂O flux in the Arctic and the important effect of light on the N₂O source and sink. However, there are some major questions. Firstly, how to make sure that the N₂O source or sink is caused by light, instead of other factors. Although the authors applied opaque chamber and nighttime measurements, the different N₂O fluxes between light and dark conditions could be mainly attributed to the temperature, moisture, etc, instead of light. Importantly, we did not observe the considerable effect of PAR on the N₂O fluxes ($R^2=0.06$) and in the random forest analysis. Secondly, the sampling methodology is unclear. Considering this study used the AERIS MIRA Ultra N₂O/CO₂ analyser, how long does it take for each measurement? Did the machine capture the N₂O flux, because it is very low in the field? What's the raw data about the relationship between N₂O concentration and time? In addition, the statistical analysis needs to improve to support the statement that N₂O source or sink depends on the light. Thirdly, the result showed that the N₂O source/sink function shifted from light to dark conditions. It might not be owing to the light, but attributed to the sampling time (daytime vs. nighttime). Furthermore, the underlying mechanism was not well investigated. The microbial characteristics, essential plant traits, and soil organic carbon are not measured. Lastly, it would be better to apply the result to the whole Arctic area.

Minor comments:

1. L124-130 To explain the effect of NH₄⁺ availability, it would be better to add the regression analysis between NH₄ and N₂O fluxes in Figure S3.
2. L195-198 It would be better to add a figure to show the temporal variation of N₂O flux from daytime to nighttime.
3. L201-203 Even if the authors conducted the measurement during the nighttime, there are other factors that interfere with the light effect, such as temperature, moisture, wind, etc. The variation of them is not entirely caused by light.
4. In the methodology section, why not use random forest analysis on the whole data? Like in the light and dark conditions.
5. L299-300 In Figure S3, the CO₂ flux seems to be the essential factor.

Reviewer #3 (Remarks to the Author):

General Comments

This study makes a substantial contribution to the field by providing an important field-measured dataset that advances our understanding of subarctic N₂O flux dynamics. The manuscript would be further strengthened if this key contribution were made more explicit and prominent, rather than being somewhat obscured by the complex discussion of multiple interacting factors, where clear directional interpretations are sometimes difficult to follow.

The primary novel data contribution of this study lies in its experimental testing of light versus dark conditions using custom-made chamber equipment. The authors conducted additional measurements to ensure that the observed effects genuinely represent dark conditions, including artificial darkening during daytime measurements and comparisons with naturally dark nighttime conditions. I suggest placing greater emphasis on this field-measurement design and rationale, as it directly explains how the new findings were generated. This emphasis may be more effective than beginning with microsite-based summaries, for example.

In other words, while the discussion of variability across microsites is valid and interesting, similar insights have been reported in previous studies. In contrast, the explicit consideration of light versus dark conditions, and the detailed look of hot spot conditions in comparison with other sites represent more novel contributions of this work with methodologically robust field-measurement framework in my opinion. This aspect could be highlighted as a methodological and conceptual advance, highlighting the gap in our current understanding of cold region soil GHG potentials with implications for future research.

In this regard, I suggest including a summary schematic or conceptual figure illustrating the measurement approach, and/or clearly explaining in the Introduction how PAR effects are tested in this study, in comparison with previous studies that have only inferred potential light effects from more limited datasets. While I recognize that the journal places the Methods section at the end, an early and clear framing of this experimental logic would greatly improve accessibility.

In addition, Arctic and subarctic systems differ in ecosystem structure and characteristics. The title refers to subarctic, whereas the abstract uses arctic throughout. It would be helpful to clarify and maintain consistency regarding the focal region of this study. Similarly, clearer and more consistent use of terminology would strengthen the manuscript, particularly for terms such as permafrost (potential permafrost zone versus the permafrost system itself), peatland (a wetland ecosystem accumulating peat), permafrost-supported peatlands (peatlands formed due to poor drainage associated with underlying permafrost), and descriptors such as nutrient-poor.

I provide specific comments below, though these are not comprehensive as I suggest the above major structural reorganization. After the manuscript undergoes major revision, I would be happy to provide more detailed follow-up comments.

Abstract

L7-9: This sentence accurately reflects the findings, but describing the response as “complex” does not add much interpretive value. It would be more informative to describe the directional responses to the investigated drivers, for example specifying how combinations of high PAR, CO₂ fluxes, and soil moisture (rather than “other factors”) influenced N₂O fluxes, and clarifying whether PAR is dominant but not exclusive.

L9-10: If “dark conditions” are intended to indicate the absence of PAR, this could be stated more explicitly. For clarity, consider phrasing with a directional comparison, for example: “the presence of PAR (light) promotes overall N₂O uptake, whereas the absence of PAR (dark conditions) ...”

Introduction

1st paragraph: The main message of this paragraph is not entirely clear. Consider clarifying the central motivation or framing question introduced here.

L16-18: Given all, why is N₂O only the third most important greenhouse gas? If this is meant for a broad audience, it may help to briefly explain that N₂O is a strong greenhouse gas but occurs at much lower atmospheric concentrations than CO₂.

L23-24: “Shifting a focus” from what? No earlier focus is explicitly stated. You might instead consider wording such as: “... high N₂O emissions from organic- and ice-rich soils in northern permafrost regions.”

L58-59: The phrase “their ecological importance” is unclear. Do you mean the potential importance of N₂O sinks or dynamics for the global nitrogen cycle, which is important for the global ecosystem productivity? Or our attempts to understand this complexity to improve our understanding of the nature (ecology as a study discipline). Also, in the phrase “Arctic N₂O sinks and are...”, the sentence appears incomplete.

L67-80: This appears to be one of the most important parts of the Introduction. It may help to briefly explain how Stewart et al. (2012) approached PAR effects on N₂O fluxes, particularly whether then the closed chambers were used. Also, it is a bit sudden to read about the temperate tree and agricultural soil measurements if this part was meant to recognize the gap in measuring the light condition N₂O fluxes in Arctic regions.

L94-98: This section reads more like a concluding remark or future research suggestion than background. It may not be necessary in the Introduction, and its connection to the subsequent discussion of thawing southern permafrost peatlands is unclear.

L106-110: Consider stating hypotheses here rather than presenting findings. While this may depend on journal style, the current phrasing feels somewhat rushed and may discourage readers before they reach the Results.

Results and Discussion

Overall, the manuscript would benefit from more clearly outlining its unique contribution to light-condition measurements of arctic and/or subarctic N₂O fluxes, especially given the scarcity of such data. One possible approach would be to reverse the order of the current discussion: begin by explaining how the field-measurement approach differs from previous studies (e.g., referencing recent reviews), then summarize earlier evidence for N₂O sinks based on limited data, and finally present the comprehensive dataset from this study showing overall N₂O uptake. This structure would better highlight how the present results advance understanding of nitrogen cycling in this critical region.

L119: Palsa lichen (PL) and L122: Palsa moss (PM) and please also define these abbreviations in the Figure 1 caption. Using these abbreviations consistently throughout the text may improve readability (at the authors' discretion).

L126-130: Please refer to Figure S3 more explicitly. Is this section emphasizing differences between PM and PL due to vegetation cover despite geomorphic similarity, or differences related to NH₄⁺ availability? If the former, you might simply note the observed difference without implying a fully resolved mechanism. If the latter, the text currently emphasizes

seasonal variation in PM relative to fens and bogs, but does not clearly distinguish PM from PL.

L136-137: Is the N₂O sink measured here smaller than that reported in other environments? Also, please clarify what is meant by "ecologically relevant." Does this refer to microbial ecosystem processes or to the broader regional or global significance for nitrogen cycling and carbon fluxes?

L207-210: The Figure 3 caption states that panels a, b, and c represent the maximum N₂O flux rate, half-saturation PAR point, and intercept, respectively. However, panel a appears to show only p-values rather than estimated values. If this is intentional, please revise the caption accordingly.

L221-238: In several places, including this paragraph, it is difficult to distinguish between observations derived from this study and inferences drawn from previous literature, as these are mixed. For example, this paragraph discusses potential mechanisms underlying the observed PAR-N₂O relationship, and the discussion of shrub effects in Palsa moss and fen sites (up to L233) sounds relatively clear, but the subsequent shift to the mechanistic difference of immediate light-dark transitions is somewhat confusing. Please check my understanding that the implications here are meant to distinguish (i) the longer duration of 'day' effect that includes PAR and other environmental factors such as soil temperatures and plant photosynthesis from (ii) the immediate effects of PAR on-off?

I think it might be helpful to clarify this point by not intermixing the day-night differences in biological effects (e.g., root exudation, plant activity) with instantaneous PAR limitation effects. For example, it may help to (i) show daytime and nighttime light-dark measurements together for the same plots in Figure S4 if possible (to show clearly the immediate dark and night conditions are more similar than the day-light conditions?), and (ii) focus this current paragraph primarily on plant-mediated responses to day-light via photosynthesis and carbon use, which isn't possible to fully confirm by the current study without long-term measurements.

Microbial interactions, including the possibility of Sphagnum-associated microbes (L235-236), may be clearer if addressed in a separate, as in the subsequent paragraph about wetter fen conditions and potential of cyanobacteria contributions (but is this really?).

In addition, Figure S5 appears to have an incomplete caption.

** Visit Nature Portfolio's author and referees' website at www.nature.com/authors for information about policies, services and author benefits**

Communications Earth & Environment is committed to improving transparency in authorship. As part of our efforts in this direction, we are now requesting that all authors identified as 'corresponding author' create and link their Open Researcher and Contributor Identifier (ORCID) with their account on the Manuscript Tracking System prior to acceptance. ORCID helps the scientific community achieve unambiguous attribution of all scholarly contributions. You can create and link your ORCID from the home page of the Manuscript Tracking System by clicking on 'Modify my Springer Nature account' and following the instructions in the link below. Please also inform all co-authors that they can add their ORCIDs to their accounts and that they must do so prior to acceptance.

Version 1:

Decision Letter:

Dear Dr Triches,

Your manuscript titled "Between light and dark, source and sink: N₂O dynamics in a subarctic, nutrient-poor permafrost peatland" has now been seen by our reviewers, whose comments appear below. In light of their advice we are delighted to say that we are happy, in principle, to publish a suitably revised version in Communications Earth & Environment.

We therefore invite you to revise your paper one last time to address the remaining concerns of our reviewers. At the same time we ask that you edit your manuscript to comply with our format requirements and to maximise the accessibility and therefore the impact of your work.

EDITORIAL REQUESTS:

Please review our specific editorial comments and requests regarding your manuscript in the attached "Editorial Requests

Table".

****Please take care to match our formatting and policy requirements. We will check revised manuscript and return manuscripts that do not comply. Such requests will lead to delays. ****

SUBMISSION INFORMATION:

OPEN ACCESS:

Communications Earth & Environment is a fully open access journal. Articles are made freely accessible on publication. For further information about article processing charges, open access funding, and advice and support from Nature Portfolio, please visit <https://www.nature.com/commsenv/open-access>

Link Redacted

Best regards,

Somaparna Ghosh, PhD
Associate Editor,
Communications Earth & Environment
Consulting Editor,
Communications Sustainability

REVIEWERS' COMMENTS:

Reviewer #1 (Remarks to the Author):

The author's revisions have improved the manuscript and addressed each of the reviewers' comments. The following are the comments that need to be addressed in the current manuscript:

L260.262... When the median is chosen as the measure of central tendency, it is statistically appropriate to report the interquartile range (IQR) or the range as the corresponding measure of dispersion, rather than the standard error or standard deviation.

Fig.2,7a. I question the use of box plots for the groups with only N=2. Generally, box plots are meaningful for sample sizes of 5 or more, as smaller samples fail to provide stable quartiles.

Reviewer #2 (Remarks to the Author):

Reviewer #2 informed us that they are happy with the revision and they suggested publication.

Reviewer #3 (Remarks to the Author):

The authors have carefully addressed the reviewers' comments and suggestions. The revised manuscript improves the clarity and emphasis of the study's main contribution, particularly the field experimental design comparing light and dark conditions. The rationale and presentation of the measurement approach are now more clearly communicated, supported by helpful visualization, and the overall structure of the manuscript has been strengthened. The revisions have also improved the consistency of terminology, enhancing the overall readability of the manuscript. I have no further comments and support publication in its current form.

** Visit Nature Portfolio's author and referees' website at www.nature.com/authors for information about policies, services and author benefits**

Response to Reviewers

1 Reviewer

I commend the author for the valuable work conducted in these regions and for highlighting the decisive role of light conditions in determining flux direction (uptake during dark periods and emission during light periods). Given the scarcity of N₂O data in these areas, this finding offers important insights for the Arctic greenhouse gas budget. Especially as the authors mentioned on Lines 136-138 ‘To the best of our knowledge, ours is the first study to report a persistent, albeit small, ecologically relevant N₂O sink in a subarctic permafrost peatland over several years, and the first to observe continuous net uptake on the dry, up-lifted palsa surfaces’. The research content is suitable for publication in *Communications Earth & Environment*. Although the large sample size from three years of greenhouse gas observations is statistically compelling, there is still a lack of in-depth mechanistic investigation in the current study. The present version has certain limitations. I have provided some comments below, which I hope the author will find useful when revising the manuscript.

We thank reviewer 1 very much for their review and insightful comments. We address their specific comments below.

Comments to Authors

1. In the Abstract section, the author directly emphasizes light as the driving force of N₂O flux. I have some doubts about the credibility of this claim. I do not deny that light may have an impact on N₂O flux, but this impact is likely not due to light directly affecting the N₂O production process. Instead, it may indirectly affect N₂O flux by influencing plant physiological processes, or it may also be due to temperature differences. However, this article omits the role of plants and temperature, directly expressing the impact of light on N₂O flux, which seems inappropriate.

Thank you for this comment. We agree that the abstract may not have put the emphasis of our study correctly: we did look at the impact of vegetation and soil temperature. We now refrain from referring to PAR as a ‘control factor’, instead provide more details about major controls during light and dark conditions. Accordingly, we have revised the abstract as follows (lines 1-16):

“Global warming and permafrost thaw in the Arctic raise concerns about increasing greenhouse gas emissions. Nitrous oxide (N₂O), a potent greenhouse gas, is produced in soils, yet its fluxes from nutrient-poor Arctic permafrost peatlands remain poorly constrained. Here, we present 1,487 chamber flux observations measured under light and dark conditions across three snow-free seasons in a sub-Arctic thawing permafrost peatland. We found a persistent and significant difference in N₂O fluxes between light (including photosynthetically active radiation, PAR) and dark (excluding PAR) conditions (Wilcoxon rank-sum test, $p < 0.001$). Under light conditions, N₂O fluxes were, on average, positive, indicating net emissions, and increased with higher photosynthesis

(more negative NEE) and PAR, and lower soil temperatures (below 3°C). In contrast, fluxes under dark conditions were consistently negative, indicating net uptake, and increased with ecosystem respiration, green canopy cover, and soil moisture. Overall, the ecosystem acted as a continuous, albeit small, N₂O sink during the snow-free season. However, we identified a persistent N₂O hotspot with substantial localized production potential. These findings highlight the importance of soil–plant–atmosphere interactions and light in regulating N₂O fluxes, with implications for Arctic greenhouse gas budgets.”

2. As mentioned in the third paragraph of the introduction, there remains an urgent need for long-term, high-resolution field observations. This study presents a three-year observation period with a substantial sample size, which is statistically robust. However, it lacks more in-depth experimental investigations into the mechanisms of N₂O, such as functional gene analysis related to N₂O production, isotopic characterization of N₂O, or tracer studies. To address these gaps while highlighting the strength of the large sample size, I recommend that the authors supplement their work by conducting a meta-analysis that aggregates global literature data from similar research areas (e.g., northern permafrost peatlands). Although collecting such data may require additional time, it would significantly enhance the global extrapolation and comparability of the study’s findings.

We thank the reviewer for their suggestion to conduct a meta-analysis to aggregate global literature data from similar research areas. We repeated our literature search, targeting specifically investigations on light effects on N₂O in pristine peatland sites, but to our surprise, did not find any. We have now modified the text to reflect this better. We also agree that further in-depth analyses would advance the field, and allow in-depth mechanical analyses of the underlying processes; however, these broader questions were out of scope for this study, since our focus is not to upscale the N₂O fluxes, but to address the drivers on a site-level. In the revised version of this manuscript, we therefore did not integrate data from similar research areas, but improved both introduction and discussion by adding more references from Arctic regions, including permafrost peatlands as follows (lines 54-78):

”The large majority of N₂O flux measurements in Arctic regions was made with opaque, closed chambers (...). However, N₂O emissions have been commonly reported to show significant diurnal variability in agricultural and forested sites, with highest emissions occurring both during day- or night-time, emphasising the importance of clarifying the PAR–N₂O relationship (1; 2). In High Arctic soils, evidence for light-dependent N₂O fluxes remains mixed. Stewart et al. (2012) observed a tendency for soils to shift from net N₂O sources in the dark to sinks under light conditions, although differences were only marginally significant ($p = 0.07$) and strongly modulated by vegetation and soil moisture (3). In contrast, Li et al. (2016) reported significantly higher N₂O emissions under light than dark conditions in tundra soils (4). Despite using comparable chamber approaches during the growing season, the two studies differ in both the magnitude and direction of the light response. Li et al. (2016) attributed enhanced N₂O emissions

under light to increased oxygen availability from photosynthesis, stimulating nitrification and coupled nitrification–denitrification in soils and plant tissues, as well as plant-mediated production or transport of N₂O from the soil (4; 5; 6). Together, these findings indicate that solar radiation can influence N₂O fluxes, but that responses are context-dependent and likely governed by interacting plant–soil processes. Interestingly, to date, this phenomenon has not been examined in pristine northern peatlands, including those affected by permafrost, although Arctic research has predominantly targeted well-drained, carbon- and nitrogen-rich permafrost features (palsas and peat plateaus) known for high N₂O emissions (7; 8). As a result, the relationship between N₂O fluxes and PAR still remain poorly understood, particularly in Arctic regions and pristine peatlands in general.”

3. We have observed that N₂O fluxes are relatively low in terrain dominated by lichens and mosses (PL/PM). Considering the close association between mosses (and to some extent lichens) and nitrogen-fixing cyanobacteria, I suggest that the discussion could further explore the impact of interactions between different functional plant types and soil microorganisms on nitrogen cycling in peatlands. Additionally, within the denitrification pathway, functional genes such as nirK (alongside others like nirS) are also involved in N₂O production and consumption. Is there any quantitative data or research on these functional genes to support this point, given that this study did not include measurements of soil microbial indicators?

Thank you for this comment. We have improved the discussion and added the impact of interactions between different vegetation types and soil microorganisms on nitrogen cycling in peatlands as follows (lines 283-295):

”Under light conditions, cyanobacteria convert N₂ to NH₃, increasing soil NH₄⁺ pools and potentially stimulating mineral N cycling, including N₂O-producing processes (9). Plant–cyanobacteria interactions may therefore play an important role in regulating nitrogen cycling in permafrost peatlands. Sphagnum mosses, which dominate anoxic environments such as bogs and fens at our study site, create acidic conditions that favour specific cyanobacterial communities while suppressing many decomposer microbes (10). In contrast, vascular plants such as sedges and shrubs can modify soil redox conditions through root oxygenation, creating microsites with different nitrogen transformation rates (11). These plant–microbe interactions generate spatial and temporal variability in nitrogen availability and may influence both nitrification and denitrification processes responsible for N₂O production (9; 12). In microsites favourable for microbial activity, rapid N turnover may lead to high gross N mineralisation, meaning that nutrients are recycled and delivered more rapidly to plants and microbes (13).” Concerning functional genes, there is a large database of microbial data (EMERGE, <https://emerge-db.asc.ohio-state.edu/>) that provides an extensive amount of microbial data for the last 15+ years, all collected at our study site. However, analysing this database is out of scope for this study.

4. I would like to highlight that different functional plant types in peatlands may have significant impacts on greenhouse gas emissions. For example, some herbaceous plants

in permafrost peatlands can access more available nitrogen through their deeper root systems. These plants may directly emit greenhouse gases by bypassing oxidation in the aerobic layer. Therefore, reporting the coverage and abundance of such plants at each observation site could help us better understand the dynamics of N₂O emissions.

Thank you very much for highlighting this. As part of our answer to this comment, we would like to refer to the answer above. We have indeed recorded the coverage of graminoids (*Eriophorum spp.* and *Carex spp.*), shrubs (predominantly *Betula nana* and *Empetrum nigrum*), lichen, peat, and moss other than peat, as well as the maximum vegetation height for peat, shrubs, and graminoids. However, even though these measurements were recorded as percentage coverage, the resolution of percentage coverage is rather coarse. Accordingly, the uncertainty associated with these numbers is high compared to the other numerical values we selected when studying the influence of potential control factors on N₂O fluxes. As a consequence, we prefer to use plant fractional coverage rather in a qualitative manner to investigate changes during the years and seasons, instead of using quantitative measurements. To still make best use of the plant information, we compromised on focussing on the green canopy cover (% of plant that is "green enough to photosynthesise"), and added these measurements to our analyses. It is, however, theoretically possible to include the data into our Random Forest models, and the results do suggest that graminoids and peat might play some role in light measurements, whereas green canopy cover is more important in dark measurements (see Figures 1, 2). However, due to the limitations of how this data was recorded and the limitations of our measurement setup and analyses (it is, e.g., not possible to disentangle the effects of photosynthesis (i.e., the uptake of CO₂ from the atmosphere) of the vegetation-driven impact on N₂O fluxes), we argue that using the green canopy cover is a more reliable way to investigate the impacts of vegetation on N₂O fluxes. Additionally, we would like to highlight that in such highly heterogeneous systems, it is challenging to disentangle the effects of vegetation from co-varying environmental factors such as soil moisture, productivity, and nutrient status. These properties vary substantially across microsites along the thaw gradient, and vegetation composition can be viewed as an integrative indicator of these underlying site conditions.

5. Lines 276-279, Plant leaf-driven N₂O fluxes primarily result from either the transport of soil-produced N₂O via photosynthesis/transpiration or are associated with phyllosphere microorganisms. Some progress has been made in understanding these processes in woody plants, which may be useful for the discussion here. Reference: Foliar N₂O emissions constitute a significant source to atmosphere. Foliar methane and nitrous oxide fluxes in *Salix bebbiana* respond to light and soil factors.

Thank you for providing this reference. We have included it as follows (lines 340-348):
"In a recent study, Karim et al. (2025) reported consistent N₂O emissions from the leaves of the pioneer tree species *Salix bebbiana* in temperate and boreal forests (14). The authors suggested that these emissions may be linked to higher photosynthetic

Figure 1: Importance of environmental variables ("features"), including vegetation cover and maximum vegetation height for light measurements.

Figure 2: Importance of environmental variables ("features"), including vegetation cover and maximum vegetation height for dark measurements.

rates that enhance nitrogen assimilation in leaves, or to increased stomatal conductance driven by changes in intercellular CO₂ concentrations, which could facilitate the release of N₂O from leaf tissues. It is possible that enhanced photosynthesis increased N assimilation and hence N₂O emissions at our site, or that increased stomatal conductance had an effect; however, more research is needed to verify the role of vegetation in emitting or transporting N₂O in Arctic ecosystems.”

6. Lines 433-434, Could the authors please introduce the light transmittance of acrylic glass?

Thank you for this comment. We changed the sentence as follows (lines 501-505):

”For our measurements, we used a custom-built static, non-steady state, non-flow-through chamber (15) made from acrylic glass (Göli GmbH, Germany, light transmittance of 92-93%) with a height of 250 mm and a diameter of 250 mm, equipped with a fan (SUNON Maglev, 80 mm x 80 mm x 25 mm, 2000 RPM) to ensure well mixed conditions within the chamber during the measurements.”

7. Lines 441-444, ‘To measure N₂O and CO₂ concentrations, we used the Aeris MIRA Ultra N₂O/CO₂ analyser (Aeris Technologies; sensitivity: 0.2 ppb/s for CO₂ and N₂O, frequency: 1 Hz); to measure CO₂ and CH₄, we used the Li-7810 CH₄/CO₂/H₂O Trace Gas Analyser from LI-COR’. I did not see the CH₄ results in this manuscript.

Thank you for pointing this out. We have removed CH₄ from the sentence and revised it as follows (lines 510-512):

”To measure N₂O and CO₂ concentrations, we used the Aeris MIRA Ultra N₂O/CO₂ analyser (Aeris Technologies; sensitivity: 0.2 ppb/s for CO₂ and N₂O, frequency: 1 Hz). ”

8. It is recommended that the proportion of invalid values removed be reported, as I would like to know how many observations were excluded.

Thank you for pointing this out. We have added the following sentence (lines 526-527):

”From our original 1654 measurement periods, 167 (10.1%) were removed through this quality-control.”

9. In Figure S1, “n=?” seems unclear. Was the sample size not reported here?

Thank you, we forgot to add the number of observations. They are now added in the Figure 3 below:

10. In Figures S6, I did not see the PAR. So what is the importance of PAR to N₂O flux?

Thank you for highlighting this, we added the revised figure for the light measurements (see Figure 4 below). Here, air temperature and CO₂ flux clearly stand out as main

Figure 3: N₂O flux over all measurement campaigns and micro habitats, excluding one hot spot, with PL, PM indicating Palsa lichen and Palsa moss, respectively. Letters indicate significance according to ANOVA and Tukey HSD post-hoc tests, with differing letters between micro habitats indicating significant differences. The dashed red horizontal line indicates the border between a source (positive values) and sink (negative values).

drivers (above 20%), and month (seasonality), green canopy cover (GCC), soil moisture, PAR, and soil temperature follow with between 5- 10% importance.

Figure 4: Importance of environmental variables ("features") **and nutrients during light measurements**, with F_CO2 = NEE, Air Temp. = air temperature, Month = measurement campaign month, Soil Moist. = soil moisture, PAR = photosynthetically active radiation, GCC = green canopy cover, Micro Hab. = micro habitat, and Soil Temp. = soil temperature

2 Reviewer

This study is interesting and shows the N₂O flux in the Arctic and the important effect of light on the N₂O source and sink. However, there are some major questions. Firstly, how to make sure that the N₂O source or sink is caused by light, instead of other factors. Although the authors applied opaque chamber and nighttime measurements, the different N₂O fluxes between light and dark conditions could be mainly attributed to the temperature, moisture, etc, instead of light. Importantly, we did not observe the considerable effect of PAR on the N₂O fluxes ($R^2=0.06$) and in the random forest analysis. Secondly, the sampling methodology is unclear. Considering this study used the Aeris MIRA Ultra N₂O/CO₂ analyser, how long does it take for each measurement? Did the machine capture the N₂O flux, because it is very low in the field? What's the raw data about the relationship between N₂O concentration and time? In addition, the statistical analysis needs to improve to support the statement that N₂O source or sink depends on the light. Thirdly, the result showed that the N₂O source/sink function shifted from light to dark conditions. It might not be owing to the light, but attributed to the sampling time (daytime vs. nighttime). Furthermore, the underlying mechanism was not well investigated. The microbial characteristics, essential plant traits, and soil organic carbon are not measured. Lastly, it would be better to apply the result to the whole Arctic area.

We thank reviewer 2 for their review and insightful comments. For their first comment (how to make sure that the N₂O flux is caused by light), we completely agree that we cannot rule out other environmental drivers, and this was also not our intention. We have rephrased and restructured our paragraphs, as well as the abstract accordingly. However, we do want to point out that the differences between light and dark measurements are significant (Wilcoxon rank-sum test 0.37, $p < 0.001$). Additionally, we would like to emphasise that our measurements periods were deliberately kept short to minimise effects of temperature and moisture changes inside the chamber (for more information, please refer to Triches et al. 2025, Practical guidelines for reproducible N₂O flux chamber measurements in nutrient-poor ecosystems. AMT, <https://amt.copernicus.org/articles/18/3407/2025/>, Figures S5 and S6). Since, in the field, we conducted our light and dark measurements within approx. 15 min, soil temperature and soil moisture did not change. The PAR effect ($R^2 = 0.06$), which is referred to, shows that the rectangular hyperbolic curve does not fit the curve of all data, and there are other environmental drivers at play, as indicated in revised manuscript as follows: (lines 251-259)

"Our results suggest that although the typical light-response curves are generally following the trend observed in all micro habitats, it was particularly N₂O fluxes from fens that followed the hyperbolic shape, suggesting that fens show the clearest diurnal cycle for N₂O fluxes driven by changing light conditions ($R^2 = 0.23$, $RMSE = 1.83$, Figure 4). While the hyperbolic shape of these curves at the fen site suggests a strong PAR effect, particularly at low PAR values, the large scatter indicates that other important drivers are also at play, particularly in the other micro habitats. According to our data, the differences between N₂O fluxes during light and dark conditions are therefore most likely a result of multiple interacting drivers."

For fens, however, the curve explains almost 25% of the variability in N_2O fluxes ($R^2 = 0.23$). Concerning the second point (unclear methodology), we have added a conceptual figure of our methods (Figure 1 in manuscript, Figure 8 below), and also indicated the chamber closure time (300 s = 5 min) in the methods. We are confident that our analyser captures very low N_2O fluxes, as we thoroughly tested this both in the laboratory and the field (for more information, please refer to: Triches et al. 2025, Practical guidelines for reproducible N_2O flux chamber measurements in nutrient-poor ecosystems. AMT, <https://amt.copernicus.org/articles/18/3407/2025/>). As it is common practice for chamber studies to report flux rates rather than concentrations, we provide our flux calculation methods, and also the raw data can be accessed on Edmond database. Concerning the third comment, we are aware that further in-depth studies are needed to explain the mechanisms behind our results, and encourage future research to focus on plant-related or/and microbial mechanisms that may explain this phenomenon. At our study site, there are restrictions on soil sampling within the footprint of an ICOS tower, which was one of the reasons why we were unable to conduct certain in-depth analyses. There has been sampling in this footprint through ICOS, but the data is not yet available. However, since we measured N_2O fluxes with transparent chambers (light) and reflective tarps (dark) during daytime, as well as during nighttime, we are confident that our results are not measurement artefacts. Lastly, we agree that a pan-Arctic upscaling of N_2O fluxes is needed, but we argue that this is out of scope for this study and would require more observations at other locations.

Figure 5: Conceptualisation of the measurement setup, with transparent (light) and opaque (dark) measurements during the day (left side), and transparent (dark) measurements during the night. Upwards arrows indicate N_2O emissions, whereas downward arrows indicate N_2O uptake into the soil.

Minor comments

1. L124-130 To explain the effect of NH_4^+ availability, it would be better to add the regression analysis between NH_4^+ and N_2O fluxes in Figure S3.

Thank you very much for this comment. We have created another figure to show the regression analysis between NH_4^+ and N_2O fluxes (see Figure 6 below). In nutrient-poor ecosystems, it is often not possible to directly link nutrients to N_2O fluxes; at our study site, this is the case for palsa lichen, bog, and fen sites. However, for palsa moss, some of the variance is indeed explained by NH_4^+ ($p = 0.004$, $R^2 = 0.12$). We have added this figure to our SI (Figure S4) and added this sentence (lines 595-596):

"However, NH_4^+ values were generally higher in palsa than bog and fen, and significantly correlated to PM sites (Figure S4)"

Figure 6: Regression analysis between measured NH_4^+ and N_2O fluxes divided into the different micro habitats (excluding the continuous hot spot), with R^2 and p-values.

2. L195-198 It would be better to add a figure to show the temporal variation of N_2O flux from daytime to nighttime.

Thank you for the comment. We have added this figure (see Figure 7 below) in the SI (Fig. S5) to show that indeed, N_2O fluxes decline during the night.

Figure 7: Diurnal cycle of N₂O fluxes from all micro habitats (excluding hot spot) during all measurement campaigns in time of day (EET). 4 measurements shortly after 24:00 were removed for improved visualisation.

3. L201-203 Even if the authors conducted the measurement during the nighttime, there are other factors that interfere with the light effect, such as temperature, moisture, wind, etc. The variation of them is not entirely caused by light.

Thank you. We agree that several environmental parameters are different during the day or night. However, as mentioned above, we conducted both light and dark measurements during the day, and these measurements were conducted within approx. 10-15 min from one another. During this time, soil temperature, soil moisture, etc. do not change, and still we see the differences in N₂O fluxes in our data. We therefore argue that even though sunlight does not seem to be the only driver of N₂O fluxes at our site, it is an important one.

4. In the methodology section, why not use random forest analysis on the whole data? Like in the light and dark conditions.

Thank you for this comment. It appears there has been a misunderstanding here. In the manuscript, we indicate, "To explore drivers of the observed N₂O fluxes, we first built several Random Forest (RF) models **on the complete dataset** to identify the most important parameters (see below)", meaning that we did start our analyses with RF analyses on the whole dataset. Since our previous text was obviously slightly misleading, we changed the sentence to (lines 546-548):

"To explore the drivers of the observed N₂O fluxes, we applied Random Forest (RF) models to the entire dataset, including all observations rather than to subsets, in order

to identify the most important predictors (see below)."

5. L299-300 In Figure S3, the CO₂ flux seems to be the essential factor.

Thank you for this comment. We agree that the CO₂ flux is the most important driver, which is why we address the relationship between CO₂ and N₂O fluxes, as well as the impact of light on these two greenhouse gases, as possible interactions (lines 334-348 and 410-421):

"During light conditions, sunlight triggers photosynthesis, leading to increased CO₂ uptake (i.e. negative NEE) during light conditions. This results in more labile C entering the soil through root exudates, which does not only boost heterotrophic activity but also drives denitrification, both directly (16) and indirectly by depleting the soil of O₂ through microbial respiration (17; 18), particularly when combined with higher soil temperatures. Plants may also release N₂O through their leaves (5), and mediate the transport of microbially produced N₂O (6; 5). In a recent study, Karim et al. (2025) reported consistent N₂O emissions from the leaves of the pioneer tree species Salix bebbiana in temperate and boreal forests (14). The authors suggested that these emissions may be linked to higher photosynthetic rates that enhance nitrogen assimilation in leaves, or to increased stomatal conductance driven by changes in intercellular CO₂ concentrations, which could facilitate the release of N₂O from leaf tissues. It is possible that enhanced photosynthesis increased N assimilation and hence N₂O emissions at our site, or that increased stomatal conductance had an effect; however, more research is needed to verify the role of vegetation in emitting or transporting N₂O in Arctic ecosystems."

"The positive correlation between N₂O and both NEE fluxes and ER emissions of the hot spot aligns with most studies on CO₂-N₂O relationships (19; 20). In an agricultural field in the UK, Alskaf et al. 2021 found a positive relationship between NEE and N₂O emissions and hypothesised that these processes are governed both below- and aboveground, i.e through soil or plant emissions (21). When, and shortly after, plants photosynthesise, they release labile organic compounds into the soil through their roots, which is expected to increase both NO₃⁻ and NH₄⁺-N in the soil (22). Since N₂O is produced from nitrate during denitrification and ammonium during nitrification (23), this leads to an increased N₂O production (22). On our site, NO₃⁻ concentrations in the surface peat were below the detection limit of the method in all micro habitats, and NH₄⁺ concentrations at the hot spot were not higher than on other palsa moss sites; however, we cannot rule out the possibility of elevated mineral N content in the deeper layers underneath the hot spots."

3 Reviewer

General Comments

This study makes a substantial contribution to the field by providing an important field-measured dataset that advances our understanding of subarctic N₂O flux dynamics. The manuscript would be further strengthened if this key contribution were made more explicit and prominent, rather than being somewhat obscured by the complex discussion of multiple interacting factors, where clear directional interpretations are sometimes difficult to follow. The primary novel data contribution of this study lies in its experimental testing of light versus dark conditions using custom-made chamber equipment. The authors conducted additional measurements to ensure that the observed effects genuinely represent dark conditions, including artificial darkening during daytime measurements and comparisons with naturally dark nighttime conditions. I suggest placing greater emphasis on this field-measurement design and rationale, as it directly explains how the new findings were generated. This emphasis may be more effective than beginning with microsite-based summaries, for example. In other words, while the discussion of variability across microsites is valid and interesting, similar insights have been reported in previous studies. In contrast, the explicit consideration of light versus dark conditions, and the detailed look of hot spot conditions in comparison with other sites represent more novel contributions of this work with methodologically robust field-measurement framework in my opinion. This aspect could be highlighted as a methodological and conceptual advance, highlighting the gap in our current understanding of cold region soil GHG potentials with implications for future research. In this regard, I suggest including a summary schematic or conceptual figure illustrating the measurement approach, and/or clearly explaining in the Introduction how PAR effects are tested in this study, in comparison with previous studies that have only inferred potential light effects from more limited datasets. While I recognize that the journal places the Methods section at the end, an early and clear framing of this experimental logic would greatly improve accessibility. In addition, Arctic and subarctic systems differ in ecosystem structure and characteristics. The title refers to subarctic, whereas the abstract uses arctic throughout. It would be helpful to clarify and maintain consistency regarding the focal region of this study. Similarly, clearer and more consistent use of terminology would strengthen the manuscript, particularly for terms such as permafrost (potential permafrost zone versus the permafrost system itself), peatland (a wetland ecosystem accumulating peat), permafrost-supported peatlands (peatlands formed due to poor drainage associated with underlying permafrost), and descriptors such as nutrient-poor.

I provide specific comments below, though these are not comprehensive as I suggest the above major structural reorganization. After the manuscript undergoes major revision, I would be happy to provide more detailed follow-up comments.

We thank reviewer 3 very much for their review and insightful comments. We are grateful for the suggestion to reorganise our manuscript and put the emphasis on our field-measurement design and rationale. We have added a conceptual figure of our methodology, our hypotheses, and made the experimental testing of light versus dark conditions more prominent. We added our specific changes in the comments below. We also improved the terminology of Arctic / sub-Arctic and peatland/palsa/etc, and thank the reviewer for point-

ing this out.

Specific Comments

Abstract

1. L7-9: This sentence accurately reflects the findings, but describing the response as “complex” does not add much interpretive value. It would be more informative to describe the directional responses to the investigated drivers, for example specifying how combinations of high PAR, CO₂ fluxes, and soil moisture (rather than “other factors”) influenced N₂O fluxes, and clarifying whether PAR is dominant but not exclusive.

Thank you for this comment. We have modified our abstract and added your suggestions as follows (lines 1-16):

”Global warming and permafrost thaw in the Arctic raise concerns about increasing greenhouse gas emissions. Nitrous oxide (N₂O), a potent greenhouse gas, is produced in soils, yet its fluxes from nutrient-poor Arctic permafrost peatlands remain poorly constrained. Here, we present 1,487 chamber flux observations measured under light and dark conditions across three snow-free seasons in a sub- Arctic thawing permafrost peatland. We found a persistent and significant difference in N₂O fluxes between light (including photosynthetically active radiation, PAR) and dark (excluding PAR) conditions (Wilcoxon rank-sum test, $p < 0.001$). Under light conditions, N₂O fluxes were, on average, positive, indicating net emissions, and increased with higher photosynthesis (more negative NEE) and PAR, and lower soil temperatures (below 3°C). In contrast, fluxes under dark conditions were consistently negative, indicating net uptake, and increased with ecosystem respiration, green canopy cover, and soil moisture. Overall, the ecosystem acted as a continuous, albeit small, N₂O sink during the snow-free season. However, we identified a persistent N₂O hotspot with substantial localized production potential. These findings highlight the importance of soil–plant–atmosphere interactions and light in regulating N₂O fluxes, with implications for Arctic greenhouse gas budgets.”

2. L9-10: If “dark conditions” are intended to indicate the absence of PAR, this could be stated more explicitly. For clarity, consider phrasing with a directional comparison, for example: “the presence of PAR (light) promotes overall N₂O uptake, whereas the absence of PAR (dark conditions) ...”

Thank you for this comment. As stated above, we have modified our abstract and have also added this suggestion as follows (lines 5-7):

”We found a persistent and significant difference in N₂O fluxes between light (including photosynthetically active radiation, PAR) and dark (excluding PAR) conditions (Wilcoxon rank-sum test, $p < 0.001$)”

Introduction

1. 1st paragraph: The main message of this paragraph is not entirely clear. Consider clarifying the central motivation or framing question introduced here.

Thank you for this comment. We have changed the 1st paragraph as follows to put the emphasis on a lack of data from nutrient-poor Arctic sites (lines 18-31):

“Nitrous oxide (N₂O) is a strong greenhouse gas with a long atmospheric lifetime (109 years) and a 298 times stronger global warming potential than the same mass of carbon dioxide (CO₂) over a time frame of 100 years- although its concentration in the atmosphere is more than thousand times lower than that of CO₂. Soils are major contributors to the global N₂O budget, both in natural and managed ecosystems (24). Arctic soils were previously considered to have a negligible impact on the global N₂O budget due to limited mineral nitrogen (N) availability resulting from slow mineralisation in cold conditions, as well as low nitrogen deposition. However, recent studies challenged this assumption by reporting high N₂O emissions from organic- and ice-rich permafrost peatlands in the northern hemisphere (25; 26; 8; 27). Because the accurate determination of low N₂O fluxes, including N₂O uptake from the atmosphere, is difficult, there has been a research bias towards high-emitting Arctic peatlands (7). Yet, nutrient-poor sites dominate the Arctic landscape (28), thus low fluxes can be important on an areal basis. Insights on N₂O fluxes from sites with limited availability of N is therefore needed to improve our understanding of the N₂O budget in the Arctic.”

2. L16-18: Given all, why is N₂O only the third most important greenhouse gas? If this is meant for a broad audience, it may help to briefly explain that N₂O is a strong greenhouse gas but occurs at much lower atmospheric concentrations than CO₂.

Thank you for this comment. We agree and have modified this as follows (lines 18-21):

“Nitrous oxide (N₂O) is a strong greenhouse gas with a long atmospheric lifetime (109 years) and a 298 times stronger global warming potential than the same mass of carbon dioxide (CO₂) over a time frame of 100 years- although its concentration in the atmosphere is more than thousand times lower than that of CO₂.”

3. L23-24: “Shifting a focus” from what? No earlier focus is explicitly stated. You might instead consider wording such as: “. . . high N₂O emissions from organic- and ice-rich soils in northern permafrost regions.”

Thank you for this comment. We have removed the wording (see full paragraph in first comment above).

4. L58-59: The phrase “their ecological importance” is unclear. Do you mean the potential importance of N₂O sinks or dynamics for the global nitrogen cycle, which is important for the global ecosystem productivity? Or our attempts to understand this complexity to improve our understanding of the nature (ecology as a study discipline). Also, in the phrase “Arctic N₂O sinks and are. . .”, the sentence appears incomplete.

Thank you for pointing this out. We have modified the sentence as follows (lines 101-103):

“Despite their potential ecological importance for the global nitrogen cycle, Arctic N₂O sinks and are not well captured in current field datasets.”

5. L67-80: This appears to be one of the most important parts of the Introduction. It may help to briefly explain how Stewart et al. (2012) approached PAR effects on N₂O fluxes, particularly whether then the closed chambers were used. Also, it is a bit sudden to read about the temperate tree and agricultural soil measurements if this part was meant to recognize the gap in measuring the light condition N₂O fluxes in Arctic regions.

Thank you for this suggestion. We have included more details about the study from Stewart et al. (2012) and included another Arctic study (Li et al. 2016) instead of referencing studies from other ecosystems. We then conclude that we did not find any studies looking at this phenomenon in pristine (permafrost) peatlands (lines 58-78):

"However, N₂O emissions have been commonly reported to show significant diurnal variability in agricultural and forested sites, with highest emissions occurring both during day- or night-time, emphasising the importance of clarifying the PAR–N₂O relationship (1; 2). In High Arctic soils, evidence for light-dependent N₂O fluxes remains mixed. Stewart et al. (2012) observed a tendency for soils to shift from net N₂O sources in the dark to sinks under light conditions, although differences were only marginally significant ($p = 0.07$) and strongly modulated by vegetation and soil moisture (3). In contrast, Li et al. (2016) reported significantly higher N₂O emissions under light than dark conditions in tundra soils (4). Despite using comparable chamber approaches during the growing season, the two studies differ in both the magnitude and direction of the light response. Li et al. (2016) attributed enhanced N₂O emissions under light to increased oxygen availability from photosynthesis, stimulating nitrification and coupled nitrification–denitrification in soils and plant tissues, as well as plant-mediated production or transport of N₂O from the soil (4; 5; 6). Together, these findings indicate that solar radiation can influence N₂O fluxes, but that responses are context-dependent and likely governed by interacting plant–soil processes. Interestingly, to date, this phenomenon has not been examined in pristine northern peatlands, including those affected by permafrost, although Arctic research has predominantly targeted well-drained, carbon- and nitrogen-rich permafrost features (palsas and peat plateaus) known for high N₂O emissions (7; 8). As a result, the relationship between N₂O fluxes and PAR still remain poorly understood, particularly in Arctic regions and pristine peatlands in general."

6. L94-98: This section reads more like a concluding remark or future research suggestion than background. It may not be necessary in the Introduction, and its connection to the subsequent discussion of thawing southern permafrost peatlands is unclear.

Thank you for this comment. We have modified the paragraph to add more details about our hypotheses and study design as follows (lines 107-124):

"To advance understanding of Arctic N₂O sinks and low N₂O effluxes, we investigated the drivers of N₂O fluxes in a thawing permafrost peatland in (sub-)Arctic Sweden under both light and dark conditions across a dry-to-wet thaw gradient from palsa to bog to fen, representing contrasting nutrient statuses and vegetation communities. At our site, palsas are dominated by lichen, shrubs, and mosses; we classify them as Palsa Lichen (PL) when dominated by lichen only, and Palsa Moss (PM) when vegetated by

shrubs, mosses, and lichens. Bogs and fens are characterized by peat-forming mosses and graminoids. According to the literature above, we expected to find 1) an N₂O sink in the wet parts of the permafrost peatland (29; 7; 30), 2) significant differences between light and dark measurements of N₂O fluxes (3; 4), and 3) higher, but still low, N₂O fluxes during the warmer summer month July compared to May and September due to higher soil temperature (31; 32; 33). To test these hypotheses, we measured N₂O and CO₂ fluxes with manual flux chambers and a portable gas analyser, and both in- and excluded sunlight by using transparent chambers (light conditions) and a light-reflective tarp (dark conditions) (34). Our measurements were taken over three years (2022-2024) in the snow-free season (n = 1487) between May and September, a period coinciding with the warmest summer in Fennoscandia in 2000 years (35). We specifically separated our analyses in light and dark flux measurements to identify if flux drivers and patterns differ in varying light conditions.”

7. L106-110: Consider stating hypotheses here rather than presenting findings. While this may depend on journal style, the current phrasing feels somewhat rushed and may discourage readers before they reach the Results.

Thank you for suggesting this. We have added our hypotheses and removed the findings summary (please refer to the comment above).

Results and Discussion

Overall, the manuscript would benefit from more clearly outlining its unique contribution to light-condition measurements of arctic and/or subarctic N₂O fluxes, especially given the scarcity of such data. One possible approach would be to reverse the order of the current discussion: begin by explaining how the field-measurement approach differs from previous studies (e.g., referencing recent reviews), then summarize earlier evidence for N₂O sinks based on limited data, and finally present the comprehensive dataset from this study showing overall N₂O uptake. This structure would better highlight how the present results advance understanding of nitrogen cycling in this critical region.

Thank you very much for suggesting this order. We have revised the order of our findings: first, we explain how our field-measurement approach differs from previous studies, then we summarise earlier evidence for N₂O sinks based on the limited data available, give our sink values, and compare them. Then, we report our findings on flux drivers of N₂O (starting with the light-dark differences), before we introduce the hot spot findings. We argue that it is more helpful for the reader to read about the previous findings of N₂O sinks (literature) in the same paragraph in which we present our findings. The modified paragraph / chapter on the N₂O sink will be given in Q. 3. We now start the results section with the following paragraph (lines 126-141):

”Compared to previous studies, our study has three main advantages: first, with our measurement set-up, we are confident to be able to measure very low N₂O flux rates (34); second, we conducted seven field campaigns (totalling to approx. five months in the field), providing us with the most extensive dataset of N₂O fluxes measured by the chamber method in Arctic regions (n = 1462, both light and dark measurements), and, as a result, third,

the largest dataset with N₂O fluxes measured in both light and dark conditions, including night measurements. Our measurement setup is comparable with previous studies but we explicitly tested the impact of the light by comparing N₂O fluxes under the presence (light) and absence (dark) of solar radiation (including PAR). For this purpose, we used a transparent chamber during daytime measurements to allow PAR exposure, and conducted additional measurements under dark conditions during the day and at night (Figure 8). To the best of our knowledge, our study is the first to do in-situ transparent chamber measurements in a pristine (sub-) Arctic permafrost peatland, confirming previous results from High Arctic soils suggesting that the impact of sunlight on N₂O fluxes may be more important in Arctic ecosystems than previously thought. These findings have significant implications for pan-Arctic N₂O budget estimates.”

Figure 8: Conceptualisation of the measurement setup, with transparent (light) and opaque (dark) measurements during the day (left side), and transparent (dark) measurements during the night. Upwards arrows indicate N₂O emissions, whereas downward arrows indicate N₂O uptake into the soil.

1. L119: Palsa lichen (PL) and L122: Palsa moss (PM) and please also define these abbreviations in the Figure 1 caption. Using these abbreviations consistently throughout the text may improve readability (at the authors’ discretion).

Thank you for pointing this out, we have added the abbreviations to the figure caption as follows. We also use the abbreviations consistently throughout the text, or simply refer to palsas.

”Mean \pm SE N₂O fluxes divided into months and micro habitats, excluding one hot spot (light and dark conditions combined), with PL and PM standing for Palsa Lichen and Palsa Moss, respectively”.

2. L126-130: Please refer to Figure S3 more explicitly. Is this section emphasizing differences between PM and PL due to vegetation cover despite geomorphic similarity, or differences related to NH_4^+ availability? If the former, you might simply note the observed difference without implying a fully resolved mechanism. If the latter, the text currently emphasizes seasonal variation in PM relative to fens and bogs, but does not clearly distinguish PM from PL.

Thank you for this comment. We agree that the section was unclear and meant the differences related to NH_4^+ . We have rephrased as follows (lines 164-168):

"We also observed a pronounced seasonal course in our N_2O measurements (Figure 2), with the sink strength peaking for all micro habitats in the warm summer months June and July. This seasonal pattern could be explained by NH_4^+ availability: in fens and bogs, NH_4^+ was highest in the early growing season, and then decreased throughout the summer (as seen in Figure S3), likely due to N uptake by plants, decreasing the soil mineral N pool."

3. L136-137: Is the N_2O sink measured here smaller than that reported in other environments? Also, please clarify what is meant by "ecologically relevant." Does this refer to microbial ecosystem processes or to the broader regional or global significance for nitrogen cycling and carbon fluxes?

Thank you for the comment, this is an important point. We have put greater emphasis to compare our values to previous Arctic studies in peatlands and wetlands, and tried to explain the discrepancies. We have also removed the words "ecologically relevant", since they are unclear. To address all points, we have added the following (lines 153-163 and 173-181):

"In contrast to the review, our flux measurements reveal a sink, whereas the review reported a source for peatlands (bogs, fens, peat plateaus, palsas) with median (25-75 percentiles): 2.5 (0.75 , 20.04) $\mu\text{g N}_2\text{O-N m}^{-2} \text{ h}^{-1}$ and mean of $24.83 \pm 61.79 \pm 0.24$ $\mu\text{g N}_2\text{O-N m}^{-2} \text{ h}^{-1}$ ($n = 30$) (7). The 30 peatland observations in the recent review are strongly biased toward high N_2O emissions, as eight originate from bare palsa surfaces where well-drained conditions, uplifted permafrost, and erosion of the ombrotrophic peat expose nutrient-rich deeper peat, while fen and bog sites, and observations at higher water-filled pore space ($\approx 60\%$), are under-represented. Our values were also lower than the previously reported sink for wetlands other than peatlands (i.e., marshes and swamps; median (25-75 percentiles): 0.42 (-0.33 , 4.83) $\mu\text{g N}_2\text{O-N m}^{-2} \text{ h}^{-1}$ ($n = 25$)), even when we included the hot spot." (...)

"The discrepancy of the recent review and our results is likely due to methodological differences, and the amount of observations considered. Using a portable gas analyser, we employed short chamber closures (5 min compared to the typical 30-60 min) (36), which are critical for detecting N_2O uptake. Under diffusion-limited conditions, N_2O is rapidly depleted from the chamber headspace, so longer closures can underestimate uptake rates (34). Our brief closure time therefore likely enabled detection of the palsa sink that earlier studies may have missed, enhancing the overall N_2O sink observed here. However, since former studies have relied exclusively on opaque chambers, they

may underestimate N₂O emissions, especially in shoulder seasons, during which measurements are rarely conducted.” (...) “Together, the methodological differences and unexplained palsa sink highlight that current permafrost peatland N₂O budgets may still be affected by measurement constraints, including both underestimated uptake due to long chamber closures, the limited amount of N₂O flux data, and incomplete representation of light-driven flux dynamics.”

4. L207-210: The Figure 3 caption states that panels a, b, and c represent the maximum N₂O flux rate, half-saturation PAR point, and intercept, respectively. However, panel a appears to show only p-values rather than estimated values. If this is intentional, please revise the caption accordingly.

Thank you for pointing this out. We have adapted this as follows:

“The curves were fitted using a non-linear least squares model, with the parameters a, b, and c representing the p-values of maximum rate of N₂O flux, the half-saturation point of PAR, and the intercept, respectively, with NS meaning “Not Significant”.”

5. L221-238: In several places, including this paragraph, it is difficult to distinguish between observations derived from this study and inferences drawn from previous literature, as these are mixed. For example, this paragraph discusses potential mechanisms underlying the observed PAR-N₂O relationship, and the discussion of shrub effects in Palsa moss and fen sites (up to L233) sounds relatively clear, but the subsequent shift to the mechanistic difference of immediate light-dark transitions is somewhat confusing. Please check my understanding that the implications here are meant to distinguish (i) the longer duration of ‘day’ effect that includes PAR and other environmental factors such as soil temperatures and plant photosynthesis from (ii) the immediate effects of PAR on-off? I think it might be helpful to clarify this point by not intermixing the day-night differences in biological effects (e.g., root exudation, plant activity) with instantaneous PAR limitation effects. For example, it may help to (i) show daytime and nighttime light-dark measurements together for the same plots in Figure S4 if possible (to show clearly the immediate dark and night conditions are more similar than the day-light conditions?), and (ii) focus this current paragraph primarily on plant-mediated responses to day-light via photosynthesis and carbon use, which isn’t possible to fully confirm by the current study without long-term measurements.

Thank you very much for this comment. We have separated these effects in two paragraphs: one focussing on vegetation and possible use of light for photosynthesis (see below) and thus the non-instantaneous impact, the other focussing on instant reactions possibly caused by cyanobacteria (please refer to the answer to your next question). We have also added some of the information to the chapter on other N₂O drivers. Firstly, in the PAR chapter, we note our vegetation observation (lines 261-271):

“The observed PAR effect can originate from three sources: plants, microbes, or plant–microbe interactions, such as labile C input from plant roots stimulating microbial activity (2). In our study, the similarity in light–dark flux differences between vegetated palsas (PM) and fen sites may be explained by higher green canopy cover (GCC) and associated photosynthetic activity during the peak growing season. This is likely due to shrubs

on palsas sites tolerating heat and drought better than *Sphagnum* spp. mosses on bogs, which tend to dry out and turn yellow (Figure S2). As a result, shrubs and cotton grass on vegetated palsas and fens may exhibit higher light-use efficiency (LUE) under low PAR — consistent with findings in Arctic tundra ecosystems (37; 38; 39). This higher LUE is reflected in similar trends for GCC (Figure 5) and may enable greater N₂O uptake in dark conditions, when low PAR becomes more limiting (Figure 3).”

In the chapter on other N₂O drivers, we have tried to combine the interacting effects of vegetation, CO₂ and N₂O fluxes (lines 334-348):

”During light conditions, sunlight triggers photosynthesis, leading to increased CO₂ uptake (i.e. negative NEE) during light conditions. This results in more labile C entering the soil through root exudates, which does not only boost heterotrophic activity but also drives denitrification, both directly (16) and indirectly by depleting the soil of O₂ through microbial respiration (17; 18), particularly when combined with higher soil temperatures. Plants may also release N₂O through their leaves (5), and mediate the transport of microbially produced N₂O (6; 5). In a recent study, Karim et al. (2025) reported consistent N₂O emissions from the leaves of the pioneer tree species *Salix bebbiana* in temperate and boreal forests (14). The authors suggested that these emissions may be linked to higher photosynthetic rates that enhance nitrogen assimilation in leaves, or to increased stomatal conductance driven by changes in intercellular CO₂ concentrations, which could facilitate the release of N₂O from leaf tissues. It is possible that enhanced photosynthesis increased N assimilation and hence N₂O emissions at our site, or that increased stomatal conductance had an effect; however, more research is needed to verify the role of vegetation in emitting or transporting N₂O in Arctic ecosystems.”

6. Microbial interactions, including the possibility of *Sphagnum*-associated microbes (L235-236), may be clearer if addressed in a separate, as in the subsequent paragraph about wetter fen conditions and potential of cyanobacteria contributions (but is this really?).

Thank you for the comment. We have added this in a new paragraph and continue as follows, aiming to be clear with our study vs other studies (lines 273-304):

”Another potential explanation for the light–dark differences in N₂O fluxes is biological N₂ fixation, the main nitrogen input at our study site (40; 41). The nitrogenase enzyme responsible for N₂ fixation is highly light-sensitive and responds rapidly to changes in light conditions (42); thus, it provides a plausible explanation for the immediate changes in N₂O flux rate when switching from light to dark chamber measurements. Biotic N₂ fixation in pristine peatlands is primarily carried out by cyanobacteria, which associate with mosses or live as symbionts in certain lichens (41; 10). Such associations with mosses are widely reported in Arctic ecosystems; also (40; 11; 43). associations with graminoids have also been reported, although not at our site. Recent studies do, however, confirm the presence of cyanobacteria at our site (44; 45), and across the Arctic approximately 65% of lichens and 44% of bryophytes fix N₂ (12). Under light conditions, cyanobacteria convert N₂ to NH₃, increasing soil NH₄⁺ pools and potentially stimulating mineral N cycling, including N₂O-producing processes (9). Plant–cyanobacteria

interactions may therefore play an important role in regulating nitrogen cycling in permafrost peatlands. Sphagnum mosses, which dominate anoxic environments such as bogs and fens at our study site, create acidic conditions that favour specific cyanobacterial communities while suppressing many decomposer microbes (10). In contrast, vascular plants such as sedges and shrubs can modify soil redox conditions through root oxygenation, creating microsites with different nitrogen transformation rates (11). These plant–microbe interactions generate spatial and temporal variability in nitrogen availability and may influence both nitrification and denitrification processes responsible for N₂O production (9; 12). In microsites favourable for microbial activity, rapid N turnover may lead to high gross N mineralisation, meaning that nutrients are recycled and delivered more rapidly to plants and microbes (13). This mechanism may help explain our observations. N₂ fixation is strongest in wetter microhabitats such as fens (43; 11), where we observed the largest light–dark flux differences. Consistent with this, Basilier et al. 1978 reported a light-dependent linear increase in nitrogen fixation at Stordalen with seasonal peaks similar to those in our data (41). Temperature may further regulate these processes: higher temperatures correlate with increased N₂ fixation rates (43; 11), with optimal nitrogenase activity around 25°C (46). At our site, air and soil temperatures increased during summer (Figure S2), and air temperatures reached 25°C in July 2023 and 2024 (Table 1). Differences between microhabitats may also reflect varying cyanobacterial species with distinct sensitivities to temperature and moisture (46).”

7. In addition, Figure S5 appears to have an incomplete caption.

Thank you for this comment. We have changed this as follows:

”Comparison between light (orange) and dark (grey) mobile chamber measurements in the nearby palsa mire Storflakket.”

References

- [1] Shurpali, N. J. *et al.* Neglecting diurnal variations leads to uncertainties in terrestrial nitrous oxide emissions. *Scientific Reports* **6**, 25739 (2016). URL <https://www.nature.com/articles/srep25739>.
- [2] Wu, Y. *et al.* Diurnal variability in soil nitrous oxide emissions is a widespread phenomenon. *Global Change Biology* **27**, 4950–4966 (2021). URL <https://onlinelibrary.wiley.com/doi/10.1111/gcb.15791>.
- [3] Stewart, K. J., Brummell, M. E., Farrell, R. E. & Siciliano, S. D. N₂O flux from plant-soil systems in polar deserts switch between sources and sinks under different light conditions. *Soil Biology and Biochemistry* **48**, 69–77 (2012). URL <https://linkinghub.elsevier.com/retrieve/pii/S0038071712000296>.
- [4] Li, F., Zhu, R., Bao, T., Wang, Q. & Xu, H. Sunlight stimulates methane uptake and nitrous oxide emission from the High Arctic tundra. *Science of The Total Environment* **572**, 1150–1160 (2016).
- [5] Hakata, M., Takahashi, M., Zumft, W., Sakamoto, A. & Morikawa, H. Conversion of the Nitrate Nitrogen and Nitrogen Dioxide to Nitrous Oxides in Plants. *Acta Biotechnologica* **23**, 249–257 (2003). URL <https://onlinelibrary.wiley.com/doi/10.1002/abio.200390032>.
- [6] Pihlatie, M., Ambus, P., Rinne, J., Pilegaard, K. & Vesala, T. Plant-mediated nitrous oxide emissions from beech (*Fagus sylvatica*) leaves. *New Phytologist* **168**, 93–98 (2005). URL <https://nph.onlinelibrary.wiley.com/doi/10.1111/j.1469-8137.2005.01542.x>.
- [7] Voigt, C. *et al.* Nitrous oxide emissions from permafrost-affected soils. *Nature Reviews Earth & Environment* **1**, 420–434 (2020). URL <https://www.nature.com/articles/s43017-020-0063-9>.
- [8] Marushchak, M. E. *et al.* Thawing Yedoma permafrost is a neglected nitrous oxide source. *Nature Communications* **12**, 7107 (2021). URL <https://www.nature.com/articles/s41467-021-27386-2>.
- [9] Robertson, G. & Groffman, P. Nitrogen transformations. In *Soil Microbiology, Ecology and Biochemistry*, 407–438 (Elsevier, 2024).
- [10] Berg, A., Danielsson, . & Svensson, B. H. Transfer of fixed-N from N₂-fixing cyanobacteria associated with the moss *Sphagnum riparium* results in enhanced growth of the moss. *Plant and Soil* **362**, 271–278 (2013). 42951898.
- [11] Stewart, K. J., Coxson, D. & Grogan, P. Nitrogen Inputs by Associative Cyanobacteria across a Low Arctic Tundra Landscape. *Arctic, Antarctic, and Alpine Research* **43**, 267–278 (2011).

- [12] Hagge, P. *et al.* Nitrogen fixation in Arctic lichens and mosses: A survey across circumpolar subzones. *Science of The Total Environment* **999**, 180264 (2025). URL <https://linkinghub.elsevier.com/retrieve/pii/S0048969725019047>.
- [13] Diáková, K. *et al.* Variation in N₂ Fixation in Subarctic Tundra in Relation to Landscape Position and Nitrogen Pools and Fluxes. *Arctic, Antarctic, and Alpine Research* **48**, 111–125 (2016).
- [14] Karim, M. R., Halim, M. A. & Thomas, S. C. Foliar methane and nitrous oxide fluxes in *Salix bebbiana* respond to light and soil factors. *Communications Earth & Environment* **6**, 493 (2025).
- [15] Livingston, G. P. & Hutchinson, G. L. Enclosure-based measurement of trace gas exchange: Applications and sources of error. In *Biogenic Trace Gases: Measuring Emissions from Soil and Water (P.A. Matson and R.C. Harriss (Eds.))*, 14–51 (John Wiley & Sons, 1995).
- [16] Firestone, M. K. & Davidson, E. A. Microbiological Basis of NO and N₂O Production and Consumption in Soil. In *Exchange of Trace Gases between Terrestrial Ecosystems and the Atmosphere (M.O., Andreae, D.S. Schimel, and G.P. Robertson (eds.))*, 7–21 (John Wiley & Sons, New York, NY, 1989). URL https://www.researchgate.net/profile/Eric-Davidson-2/publication/246820772_Microbiological_Basis_of_NO_and_N2O_Production_and_Consumption_in_Soil/links/0c960526a6440d45a5000000/Microbiological-Basis-of-NO-and-N2O-Production-and-Consumption-in-Soil.pdf.
- [17] Blackmer, A. M., Robbins, S. G. & Bremner, J. M. Diurnal Variability in Rate of Emission of Nitrous Oxide from Soils. *Soil Science Society of America Journal* **46**, 937–942 (1982).
- [18] Farquharson, R. & Baldock, J. Concepts in modelling N₂O emissions from land use. *Plant and Soil* **309**, 147–167 (2008).
- [19] Xu, X., Tian, H. & Hui, D. Convergence in the relationship of CO₂ and N₂O exchanges between soil and atmosphere within terrestrial ecosystems. *Global Change Biology* **14**, 1651–1660 (2008). URL <https://onlinelibrary.wiley.com/doi/10.1111/j.1365-2486.2008.01595.x>.
- [20] Zona, D. *et al.* N₂O fluxes of a bio-energy poplar plantation during a two years rotation period. *GCB Bioenergy* **5**, 536–547 (2013). URL <https://onlinelibrary.wiley.com/doi/10.1111/gcbb.12019>.
- [21] Alskaf, K., Mooney, S., Sparkes, D., Wilson, P. & Sjögersten, S. Short-term impacts of different tillage practices and plant residue retention on soil physical properties and greenhouse gas emissions. *Soil and Tillage Research* **206**, 104803 (2021). URL <https://linkinghub.elsevier.com/retrieve/pii/S0167198720305857>.

- [22] Del Grosso, S. J. *et al.* General model for N₂ O and N₂ gas emissions from soils due to denitrification. *Global Biogeochemical Cycles* **14**, 1045–1060 (2000). URL <https://agupubs.onlinelibrary.wiley.com/doi/10.1029/1999GB001225>.
- [23] Hénault, C. *et al.* Predicting *in situ* soil N₂ O emission using NOE algorithm and soil database. *Global Change Biology* **11**, 115–127 (2005). URL <https://onlinelibrary.wiley.com/doi/10.1111/j.1365-2486.2004.00879.x>.
- [24] Tian, H. *et al.* Global nitrous oxide budget (1980–2020). *Earth System Science Data* **16**, 2543–2604 (2024). URL <https://essd.copernicus.org/articles/16/2543/2024/>.
- [25] Repo, M. E. *et al.* Large N₂O emissions from cryoturbated peat soil in tundra. *Nature Geoscience* **2**, 189–192 (2009). URL <https://www.nature.com/articles/ngeo434>.
- [26] Marushchak, M. E. *et al.* Hot spots for nitrous oxide emissions found in different types of permafrost peatlands: NITROUS OXIDE FLUXES FROM PERMAFROST PEATLANDS. *Global Change Biology* **17**, 2601–2614 (2011). URL <https://onlinelibrary.wiley.com/doi/10.1111/j.1365-2486.2011.02442.x>.
- [27] Elberling, B., Christiansen, H. H. & Hansen, B. U. High nitrous oxide production from thawing permafrost. *Nature Geoscience* **3**, 332–335 (2010). URL <https://www.nature.com/articles/ngeo803>.
- [28] Virkkala, A.-M. *et al.* High-resolution spatial patterns and drivers of terrestrial ecosystem carbon dioxide, methane, and nitrous oxide fluxes in the tundra. *Biogeosciences* **21**, 335–355 (2024). URL <https://bg.copernicus.org/articles/21/335/2024/>.
- [29] Martikainen, P. J., Nykänen, H., Crill, P. & Silvola, J. Effect of a lowered water table on nitrous oxide fluxes from northern peatlands. *Nature* **366**, 51–53 (1993). URL <https://www.nature.com/articles/366051a0>.
- [30] Brummell, M. E., Farrell, R. E. & Siciliano, S. D. Greenhouse gas soil production and surface fluxes at a high arctic polar oasis. *Soil Biology and Biochemistry* **52**, 1–12 (2012). URL <https://linkinghub.elsevier.com/retrieve/pii/S0038071712001290>.
- [31] Koponen, H. T., Escudé Duran, C., Maljanen, M., Hytönen, J. & Martikainen, P. J. Temperature responses of NO and N₂O emissions from boreal organic soil. *Soil Biology and Biochemistry* **38**, 1779–1787 (2006). URL <https://linkinghub.elsevier.com/retrieve/pii/S0038071706000563>.
- [32] Voigt, C. *et al.* Warming of subarctic tundra increases emissions of all three important greenhouse gases – carbon dioxide, methane, and nitrous oxide. *Global Change Biology* **23**, 3121–3138 (2017). URL <https://onlinelibrary.wiley.com/doi/abs/10.1111/gcb.13563>. Number: 8 .eprint: <https://onlinelibrary.wiley.com/doi/pdf/10.1111/gcb.13563>.

- [33] Wu, D. *et al.* The effect of nitrification inhibitor on N₂O, NO and N₂ emissions under different soil moisture levels in a permanent grassland soil. *Soil Biology and Biochemistry* **113**, 153–160 (2017). URL <https://linkinghub.elsevier.com/retrieve/pii/S0038071716306885>.
- [34] Triches, N. Y. *et al.* Practical guidelines for reproducible N₂ O flux chamber measurements in nutrient-poor ecosystems. *Atmospheric Measurement Techniques* **18**, 3407–3424 (2025). URL <https://amt.copernicus.org/articles/18/3407/2025/>.
- [35] Rantanen, M., Helama, S., Räisänen, J. & Gregow, H. Summer 2024 in northern Fennoscandia was very likely the warmest in 2000 years. *npj Climate and Atmospheric Science* **8**, 158 (2025). URL <https://www.nature.com/articles/s41612-025-01046-4>.
- [36] Schulze, C. *et al.* Nitrous Oxide Fluxes in Permafrost Peatlands Remain Negligible After Wildfire and Thermokarst Disturbance. *Journal of Geophysical Research: Biogeosciences* **128**, e2022JG007322 (2023).
- [37] Huemmrich, K. F. *et al.* Arctic Tundra Vegetation Functional Types Based on Photosynthetic Physiology and Optical Properties. *IEEE Journal of Selected Topics in Applied Earth Observations and Remote Sensing* **6**, 265–275 (2013). URL <http://ieeexplore.ieee.org/document/6507561/>.
- [38] Williams, M., Rastetter, E. B., Van der Pol, L. & Shaver, G. R. Arctic canopy photosynthetic efficiency enhanced under diffuse light, linked to a reduction in the fraction of the canopy in deep shade. *New Phytologist* **202**, 1267–1276 (2014). URL <https://onlinelibrary.wiley.com/doi/abs/10.1111/nph.12750>. Number: 4. eprint: <https://onlinelibrary.wiley.com/doi/pdf/10.1111/nph.12750>.
- [39] Pei, Y. *et al.* Evolution of light use efficiency models: Improvement, uncertainties, and implications. *Agricultural and Forest Meteorology* **317**, 108905 (2022). URL <https://linkinghub.elsevier.com/retrieve/pii/S0168192322000983>.
- [40] Granhall, U. & Selander, H. Nitrogen Fixation in a Subarctic Mire. *Oikos* **24**, 8 (1973). URL <https://www.jstor.org/stable/3543247?origin=crossref>.
- [41] Basilier, K., Granhall, U., Stenström, T.-A. & Stenstrom, T.-A. Nitrogen Fixation in Wet Minerotrophic Moss Communities of a Subarctic Mire. *Oikos* **31**, 236 (1978). URL <https://www.jstor.org/stable/3543568?origin=crossref>.
- [42] Severin, I. & Stal, L. J. Light dependency of nitrogen fixation in a coastal cyanobacterial mat. *The ISME Journal* **2**, 1077–1088 (2008). URL <https://academic.oup.com/ismej/article/2/10/1077/7588472>.
- [43] Zielke, M., Ekker, A. S., Olsen, R. A., Spjelkavik, S. & Solheim, B. The Influence of Abiotic Factors on Biological Nitrogen Fixation in Different Types of Vegetation in the High Arctic, Svalbard. *Arctic, Antarctic, and Alpine Research* **34**, 293–299 (2002).

- [44] Carrell, A. A. *et al.* Novel metabolic interactions and environmental conditions mediate the boreal peatmoss-cyanobacteria mutualism. *The ISME Journal* **16**, 1074–1085 (2022). URL <https://academic.oup.com/ismej/article/16/4/1074-1085/7474316>.
- [45] Hamard, S. *et al.* Contribution of microbial photosynthesis to peatland carbon uptake along a latitudinal gradient. *Journal of Ecology* **109**, 3424–3441 (2021). URL <https://besjournals.onlinelibrary.wiley.com/doi/10.1111/1365-2745.13732>.
- [46] Gentili, F., Nilsson, M.-C., Zackrisson, O., DeLuca, T. H. & Sellstedt, A. Physiological and molecular diversity of feather moss associative N₂-fixing cyanobacteria. *Journal of Experimental Botany* **56**, 3121–3127 (2005). URL <http://academic.oup.com/jxb/article/56/422/3121/749504/Physiological-and-molecular-diversity-of-feather>.

Response to Reviewers

We thank all three reviewers and the editor very much for the time and effort they spent on our work. Their insightful comments have greatly improved our manuscript- thank you!

1 Reviewer

The author's revisions have improved the manuscript and addressed each of the reviewers' comments. The following are the comments that need to be addressed in the current manuscript:

1. L260-262. When the median is chosen as the measure of central tendency, it is statistically appropriate to report the interquartile range (IQR) or the range as the corresponding measure of dispersion, rather than the standard error or standard deviation.

We thank the reviewer for noticing and highlighting this. We completely agree and have given the Q1-Q3 ranges (25-75 percentiles) of all our flux values:

L. 138-144: In our study, between September 2022 and August 2024, we measured median (25-75 percentiles) N₂O fluxes of -0.28 (-1.24, 0.68) $\mu\text{g N}_2\text{O-N m}^{-2} \text{h}^{-1}$ ($n = 1462$, both light and dark measurements), including one constant hot spot on a palsa with N₂O emissions of 6.08 (2.38, 21.9) $\mu\text{g N}_2\text{O-N m}^{-2} \text{h}^{-1}$ ($n = 79$; see last paragraph). When we exclude this hot spot (as done for most of the analyses), N₂O fluxes were -0.38 (-1.32, 0.49) $\mu\text{g N}_2\text{O-N m}^{-2} \text{h}^{-1}$ ($n = 1383$), indicating an N₂O sink (Figure 2 and S1).

L. 150-153: Our values were also lower than the previously reported sink for wetlands other than peatlands (i.e., marshes and swamps; median (25-75 percentiles): 0.42 (-0.33, 4.83) $\mu\text{g N}_2\text{O-N m}^{-2} \text{h}^{-1}$ ($n = 25$)), even when we included the hot spot.

L. 159-162: Overall, the vast majority of monthly mean fluxes were negative, except for net emissions observed from PL in May (median (25-75 percentiles) 0.3 (-0.49, 1.00) $\mu\text{g N}_2\text{O-N m}^{-2} \text{h}^{-1}$, $n = 162$), and from fen in September (median (25-75 percentiles) 0.11 (-1.05, 1.15) $\mu\text{g N}_2\text{O-N m}^{-2} \text{h}^{-1}$, $n = 85$).

L. 202-206: Under light conditions, fluxes were on average positive, showing net emissions (median (25-75 percentiles): 0.42 (-0.27, 1.17) $\mu\text{g N}_2\text{O-N m}^{-2} \text{h}^{-1}$ ($n = 673$, excluding hot spot)). In contrast, fluxes measured in the dark were consistently negative, indicating net uptake (median (25-75 percentiles): -1.06 (-1.06, -2.05) $\mu\text{g N}_2\text{O-N m}^{-2} \text{h}^{-1}$ ($n = 710$, excluding hot spot)), respectively (Figure 3). L. 364-367: When we include this hot spot, our study site may act as a net N₂O source, with median (25-75 percentiles) N₂O fluxes of -0.28 (-1.24, 0.68) $\mu\text{g N}_2\text{O-N m}^{-2} \text{h}^{-1}$ ($n = 1462$), and PM showing low emissions of median (25-75 percentiles) 0.01 (-1.03, 1.53) $\mu\text{g N}_2\text{O-N m}^{-2} \text{h}^{-1}$ ($n = 344$).

2. Fig.2,7a. I question the use of box plots for the groups with only N=2. Generally, box plots are meaningful for sample sizes of 5 or more, as smaller samples fail to provide stable quartiles.

We thank the reviewer for pointing this out. While we agree with the statistical reasoning, we argue that in Figure 6 (Figure 1 below), where the aim is to highlight the magnitude of the bare soil and continuous hot spot vs the low-flux micro habitats, the removal of the boxplot would not make this comparison intuitive. By using a boxplot, the magnitude differences become very clear. We have, however, added a note in the figure caption. We have also adjusted our Figure 7a (Figure 2 below) accordingly by grouping the observations into measurement months.

Figure 1: N₂O fluxes in the Stordalen mire across three snow-free seasons, including a bare hot spot (Bare) found in specific hot spot screening and the continuous vegetated hot spot (Hot spot), with PL and PM indicating Palsa Lichen and Palsa Moss, respectively. *Please note that for better visualisation, "Bare" is shown as boxplot despite low sample size (n = 2).* "Hot spot" only contains the emissions from the continuous hot spot. The red dashed line marks the border between source (positive flux values) and sink (negative flux values). Note that all N₂O fluxes measured in both light and dark conditions are shown.

Figure 2: Drivers of the single vegetated palsa moss N_2O hot spot, with a) showing light (orange boxplots) and dark (grey boxplots) measurements during all measurement campaigns and the red dashed line indicating source (positive values) and sink (negative values). Other plots show simple regression models between the N_2O flux and b) Soil temperature at 15 cm depth, c) Volumetric Water content (%), i.e. soil moisture at 12 cm depth, d) Green canopy cover, e) Active layer depth, and f) CO_2 flux. We tested linear, quadratic, logarithmic and exponential models and compared them by AIC. The lowest AIC for all predictors were given by the exponential model: $y = e^{\beta_0 + \beta_1 x}$, where y is N_2O flux and x is the predictor.